# The kinase DYRK1A reciprocally regulates the differentiation of Th17 and regulatory T cells

Bernard Khor[1,2,3,4], John D Gagnon[3], Gautam Goel[2], Marly I Roche[5], Kara L Conway[1,2,3], Khoa Tran[6], Leslie N Aldrich[3,7], Thomas B Sundberg[3], Alison M Paterson[8,9], Scott Mordecai[4], David Dombkowski[4], Melanie Schirmer[3], Pauline H Tan[6], Atul K Bhan[4], Rahul Roychoudhuri[10], Nicholas P Restifo[10], John J O'Shea[11], Benjamin D Medoff[5], Alykhan F Shamji[3], Stuart L Schreiber[3,7], Arlene H Sharpe[8,9], Stanley Y Shaw[6], Ramnik J Xavier[1,2,3]*

[1]Gastrointestinal Unit and Center for the Study of Inflammatory Bowel Disease, Massachusetts General Hospital, Harvard Medical School, Boston, United States; [2]Center for Computational and Integrative Biology, Massachusetts General Hospital, Harvard Medical School, Boston, United States; [3]Broad Institute of MIT and Harvard, Cambridge, United States; [4]Pathology Service, Massachusetts General Hospital, Boston, United States; [5]Pulmonary and Critical Care Unit, Massachusetts General Hospital, Boston, United States; [6]Center for Systems Biology, Massachusetts General Hospital, Harvard Medical School, Boston, United States; [7]Department of Chemistry and Chemical Biology, Harvard University, Cambridge, United States; [8]Department of Microbiology and Immunobiology, Harvard Medical School, Boston, United States; [9]Department of Pathology, Brigham & Women's Hospital, Harvard Medical School, Boston, United States; [10]Center for Cancer Research, National Cancer Institute, National Institutes of Health, Bethesda, United States; [11]Molecular Immunology and Inflammation Branch, National Institute of Arthritis and Musculoskeletal and Skin Diseases, National Institutes of Health, Bethesda, United States

*For correspondence: xavier@molbio.mgh.harvard.edu

Competing interests: The authors declare that no competing interests exist.

**Abstract** The balance between Th17 and T regulatory ($T_{reg}$) cells critically modulates immune homeostasis, with an inadequate $T_{reg}$ response contributing to inflammatory disease. Using an unbiased chemical biology approach, we identified a novel role for the dual specificity tyrosine-phosphorylation-regulated kinase DYRK1A in regulating this balance. Inhibition of DYRK1A enhances $T_{reg}$ differentiation and impairs Th17 differentiation without affecting known pathways of $T_{reg}$/Th17 differentiation. Thus, DYRK1A represents a novel mechanistic node at the branch point between commitment to either $T_{reg}$ or Th17 lineages. Importantly, both $T_{reg}$ cells generated using the DYRK1A inhibitor harmine and direct administration of harmine itself potently attenuate inflammation in multiple experimental models of systemic autoimmunity and mucosal inflammation. Our results identify DYRK1A as a physiologically relevant regulator of $T_{reg}$ cell differentiation and suggest a broader role for other DYRK family members in immune homeostasis. These results are discussed in the context of human diseases associated with dysregulated DYRK activity.

## Introduction

The appropriate and balanced differentiation of naïve CD4$^+$ T helper (Th) cells into either pro-inflammatory effector subsets, such as Th1 and Th17 cells, or anti-inflammatory subsets, largely

**eLife digest** Inflammation is used by the immune system to protect and repair tissues after an injury or infection. However, if inflammation is too strong, or goes on for too long, it can damage tissues. This is seen in autoimmune diseases such as inflammatory bowel disease and type 1 diabetes. Therefore, precise regulation of the inflammatory response is essential for maintaining human health.

White blood cells known as T cells are central regulators of tissue inflammation. To achieve this goal, they develop into subtypes with specialized roles. For example, some T helper cells release chemical signals that trigger inflammation and other immune responses. Regulatory T ($T_{reg}$) cells then shut down these immune responses once they are no longer needed. Many autoimmune and other inflammatory diseases are thought to arise—at least partially—because $T_{reg}$ cells fail to stop the inflammatory response. Boosting the number or the activity of $T_{reg}$ cells could therefore help to treat these diseases. However, technical difficulties have made it difficult to investigate the genes and molecular pathways that control how this subtype of white blood cells develops.

Khor et al. thought that discovering new chemicals that increase the number of $T_{reg}$ cells without harming them could help to identify the pathways that control their development. Khor et al. screened over 3000 chemicals, many of which are drugs currently approved for use in humans, for their effect on immature T cells that were taken from mice and grown in the laboratory. This 'unbiased chemical biology' approach identified several chemicals that both encouraged the T cells to develop into $T_{reg}$ cells and reduced the numbers that became inflammation-promoting T helper cells.

Khor et al. then focused on one of these chemicals, called harmine. Tests in mice showed that harmine reduces the extent of experimentally induced inflammatory reactions. $T_{reg}$ cells generated by treating immature T cells with harmine had the same effect. Further experiments showed that harmine exerts these effects, at least in part, by inhibiting the activity of a protein called DYRK1A. When DYRK1A was removed from maturing mouse T cells grown in the laboratory, the T cells tended to develop into anti-inflammatory $T_{reg}$ cells.

These findings therefore identify DYRK1A as part of a pathway that suppresses the development of $T_{reg}$ cells. It remains to be discovered how it does this, and whether other DYRK protein family members have similar roles.

represented by regulatory T ($T_{reg}$) cells, is an important determinant of immune homeostasis, dysregulation of which underlies the pathology of inflammatory diseases and cancer (*Josefowicz et al., 2012*; *Cretney et al., 2013*). Genome-wide association studies of inflammatory diseases such as type 1 diabetes (T1D) and inflammatory bowel disease (IBD) support this notion, implicating genes important for the differentiation and function of $T_{reg}$ cells (*Khor et al., 2011*). The active translational interest in manipulating this process is exemplified by recent attempts using low-dose IL-2 to enhance $T_{reg}$ cells and attenuate the inflammation associated with graft-versus-host disease and HCV vasculitis (*Koreth et al., 2011*; *Saadoun et al., 2011*; *von Boehmer and Daniel, 2012*). While these results are encouraging, IL-2 has numerous effects and reflects the larger issue that more targeted therapies to specifically manipulate individual Th lineages remain lacking, due at least in part to an incomplete understanding of the pathways that contribute to Th differentiation.

Our interest has focused on discovering novel pathways that regulate the differentiation of $T_{reg}$ cells, which represent the major anti-inflammatory Th component. The canonical pathways underlying $T_{reg}$ cell differentiation have been well described. In this regard, commitment to the $T_{reg}$ cell lineage is exemplified by expression of the hallmark transcription factor FOXP3 and occurs in either the thymus or the periphery (*Josefowicz et al., 2012*). The canonical cytokine that drives $T_{reg}$ cell differentiation is TGF-β1 and the differentiation of peripheral $T_{reg}$ ($pT_{reg}$) cells requires TGF-β1 signaling through SMAD2 and SMAD3 (*Gu et al., 2012*; *Josefowicz et al., 2012*). TGF-β1 also plays a role in thymic $T_{reg}$ ($tT_{reg}$) cell differentiation, as exemplified by the significant but transient decrease in early $T_{reg}$ cell generation upon T cell-specific deletion of the TGF-β1 RI subunit (*Liu et al., 2008*). However, TGF-β1 may be more important for maintaining the pool of $T_{reg}$ cell precursors than instructing Foxp3 expression in this compartment (*Josefowicz et al., 2012*). Similarly to IL-2, TGF-β1 exerts multiple effects on different cell types and has not proven to be a clinically useful target to

manipulate T$_{reg}$ cell differentiation, again pointing to the need to better understand the breadth of pathways involved.

In this regard, studies in SMAD2/3 doubly deficient mice point to TGF-β1-dependent, SMAD2/3-independent signals in tT$_{reg}$ cell differentiation and function, demonstrating the relevance of non-canonical pathways, even in the context of well-described cytokines (*Gu et al., 2012*). Elucidating such ancillary pathways in Th differentiation has been approached in several ways. Notable amongst these have been gene expression profiling experiments to identify differentially expressed genes. This approach has been more successful for Th17 cell differentiation, pointing to the transcription factors *Batf*, *Ahr* and *Ikzf3* and the sodium chloride sensor *Sgk1* (*Veldhoen et al., 2008*; *Schraml et al., 2009*; *Wu et al., 2013*), than for T$_{reg}$ cell differentiation. Such findings have implications for diagnostic efforts and advancing our understanding of disease pathophysiology. For example, the finding that mutations in *STAT3* (which transduces signals from IL-6, a canonical Th17 cytokine) can lead to hyper-IgE syndrome (HIES) led to the discovery that this subset of HIES patients fail to generate Th17 cells, potentially accounting for their susceptibility to fungal infection (*Ma et al., 2008*). There are also therapeutic implications; for instance, the discovery that SGK1 regulates Th17 cell differentiation led to the hypothesis that increased dietary salt intake may contribute to increased risk of autoimmune disease (*Kleinewietfeld et al., 2013*). Thus, discovering additional pathways that regulate T$_{reg}$ cell differentiation is an important effort that may benefit from other approaches.

Integrative computational analyses represent one promising adjunctive approach. Analyses of over 100 gene expression profiles of various CD4$^+$ subsets led to the discovery of novel transcription factors, including *Lef1* and *Gata1*, that regulate T$_{reg}$ cell differentiation and a model of how they cooperate to establish the T$_{reg}$ cell transcription profile (*Fu et al., 2012*). In another example, the compilation of 557 publicly available microarrays covering over 100 tissues and primary cells facilitated the discovery of *Zbtb25* as a transcription factor predominantly expressed in T cells that represses NFAT signaling in response to T cell receptor engagement (*Benita et al., 2010*). Another emerging key approach uses chemical methods to decipher novel nodes that control signal transduction pathways within T cells; this provides an important and complementary view into disease architecture by highlighting druggable connections between disease pathways less easily uncovered genetically. In this regard, defects in autophagy have been associated with IBD. Efforts to find compounds that enhance autophagy led to the observation that some autophagy-enhancing compounds specifically inhibit Th17 cell differentiation while another subset specifically enhances T$_{reg}$ cell differentiation, suggesting that these compounds highlight targets which modulate distinct sets of disease-relevant pathways (*Shaw et al., 2013*). Finally, chemoinformatic methods can help generate high-yield mechanistic hypotheses based on relevant compounds identified by chemical biology approaches. For instance, the use of chemoinformatics to predict novel binding targets for clinically used drugs based on structural similarity to other compounds that bind said targets has helped predict mechanistic explanations for clinically observed side effects (*Keiser et al., 2007*; *Lounkine et al., 2012*). Of note, these approaches are not mutually exclusive, but rather are expected to be synergistic.

Supporting the value of a chemical biology approach, compounds previously identified to modulate T$_{reg}$ cell differentiation have provided important insights into relevant signaling modules. For example, mechanistic studies of all-*trans* retinoic acid (ATRA) and rapamycin, two well-studied T$_{reg}$ cell enhancers, pointed to roles for RAR-α and mTOR signaling in T$_{reg}$ cell differentiation respectively (*Coombes et al., 2007*; *Mucida et al., 2007*; *Sun et al., 2007*; *Haxhinasto et al., 2008*; *Hill et al., 2008*; *Sauer et al., 2008*; *Hall et al., 2011*). More recently, the discovery of the microbial metabolites proprionate and butyrate as enhancers of T$_{reg}$ cell differentiation, amongst other effects, have highlighted roles for the short-chain fatty acid receptor GPR43 and histone deacetylases in T$_{reg}$ cell differentiation (*Arpaia et al., 2013*; *Furusawa et al., 2013*; *Smith et al., 2013*). These studies highlight several SMAD-distinct signals in T$_{reg}$ cell differentiation and illustrate how the discovery of novel molecules can facilitate a deeper understanding of the underlying mechanisms and pathways affecting T$_{reg}$ cell differentiation.

Th differentiation is a complex cellular process for which there exists no good simplified substitute assay. Chemical biology approaches to study this process have typically either maintained the complexity of the system (i.e., used primary CD4$^+$ T cells) to study one or two selected compounds, for example ATRA, or used larger chemical libraries to interrogate a highly simplified system. Here, we

take the novel approach of applying unbiased chemical biology to primary CD4$^+$ T cells in order to discover novel regulators of T$_{reg}$ cell differentiation. We report 14 novel compounds that specifically enhance the differentiation of T$_{reg}$ cells, but of neither Th1 nor Th17 cells. In particular, the β-carboline alkaloid harmine enhances the differentiation of T$_{reg}$ cells and potently inhibits Th17 cell differentiation, at least in part by inhibiting the activity of the kinase DYRK1A. Importantly, we demonstrate that harmine-enhanced T$_{reg}$ cells retain normal suppressive function in vitro and attenuate disease in experimental models of systemic autoimmunity and mucosal inflammation in two distinct compartments. Notably, direct administration of harmine attenuates airway inflammation in an experimental model of asthma. Our approach exemplifies how chemical biology can be applied to a physiologically relevant experimental system with a functional readout to identify DYRKs as a novel and druggable pathway that impacts T$_{reg}$ cell differentiation.

## Results

### Identifying novel small molecule enhancers of T$_{reg}$ cell differentiation

We hypothesized that identifying novel compounds that enhance T$_{reg}$ cell differentiation would enable the discovery of novel pathways that regulate this process. Accordingly, we designed an experimental workflow that feeds into an integrative computational analysis pipeline to identify small molecules that specifically enhance differentiation of T$_{reg}$ cells, but not pro-inflammatory lineages, highlight putative mechanistic classes and demonstrate functional relevance of prioritized small molecule(s) (*Figure 1A*). Primary murine CD4$^+$ T cells were reproducibly differentiated into T$_{reg}$, Th1 or Th17 lineages in a manner dependent upon the concentration of TGF-β1, IL-12 or IL-6 and/or IL-1β respectively (*Figure 1—figure supplement 1*). Lineage commitment was determined using the gold standard of flow cytometric detection of FOXP3, IFNγ and/or IL-17. The role of these canonical cytokines has been well described; the positive control for each lineage was high concentrations of lineage-driving cytokines consistent with published literature while negative controls included cells cultured under Th0 conditions without any lineage-promoting cytokines, as well as cells driven to other lineages (e.g., for T$_{reg}$ cells, negative controls were Th0, Th1 and Th17 conditions). Conditions driving the differentiation of sub- or near-maximal levels (typically about 30% and 95% of maximal levels, respectively) of T$_{reg}$, Th1 and Th17 cells (hereafter T$_{reg}$$^{low}$, Th1$^{low}$, Th17$^{low}$, T$_{reg}$$^{hi}$, Th1$^{hi}$ and Th17$^{hi}$ conditions) were identified (*Figure 1—figure supplement 1*, blue and red arrowheads respectively). Addition of a compound to sub-maximal conditions allows quantitation of its ability to enhance lineage-specific differentiation, while addition to near-maximal conditions allows quantitation of its inhibitory effect. To facilitate comparisons between experiments, we used the fractional enhancement (Fr enhance) metric, where the compound-driven difference in Th differentiation is normalized against the difference between the positive and negative controls within the experiment (i.e., Th$^{hi}$ − Th$^{low}$).

To find small molecules that enhance T$_{reg}$ cell differentiation, T$_{reg}$$^{low}$ conditions were used to screen 3281 compounds comprising FDA-approved drugs and tool compounds with known mechanisms (*Shaw et al., 2013*). Our studies revealed a previously unreported negative correlation between the number of live cells in culture and the percentage of FOXP3$^+$ cells in T$_{reg}$$^{low}$ conditions, not observed in T$_{reg}$$^{hi}$, Th1$^{low/hi}$ or Th17$^{low/hi}$ conditions (*Figure 1—figure supplement 2*). Because the compounds tested exhibit variable effects on cellularity, we accounted for corresponding effects on T$_{reg}$ cell differentiation using a linear regression model (*Figure 1—figure supplement 3*). Numerous compounds previously reported to enhance T$_{reg}$ cell differentiation were recovered, including the hypolipidemic statins (lovastatin and simvastatin), artemisinin and ATRA as well as related retinoic acids (9-*cis* retinoic acid and 13-*cis* retinoic acid), validating our experimental approach (*Coombes et al., 2007*; *Mucida et al., 2007*; *Sun et al., 2007*; *Kagami et al., 2009*; *Kim et al., 2010*; *Zhao et al., 2012*). The fractional enhancement of the weakest of these known enhancers (artemisinin, 0.3) was used as a minimum threshold to find all compounds that enhance T$_{reg}$ cell differentiation at least as strongly. By this criterion, 70 compounds were selected for retesting.

### Finding compounds that specifically enhance T$_{reg}$ cell differentiation

The compounds prioritized by our efforts described above were retested (using T$_{reg}$$^{low}$ conditions) at 8 doses that typically covered over a 1000-fold difference in concentration, allowing us to capture more optimal concentrations at which a compound might work (*Figure 1B*). In order to filter for compounds that only enhance T$_{reg}$ cell differentiation, these compounds were also tested under Th1$^{low}$ and

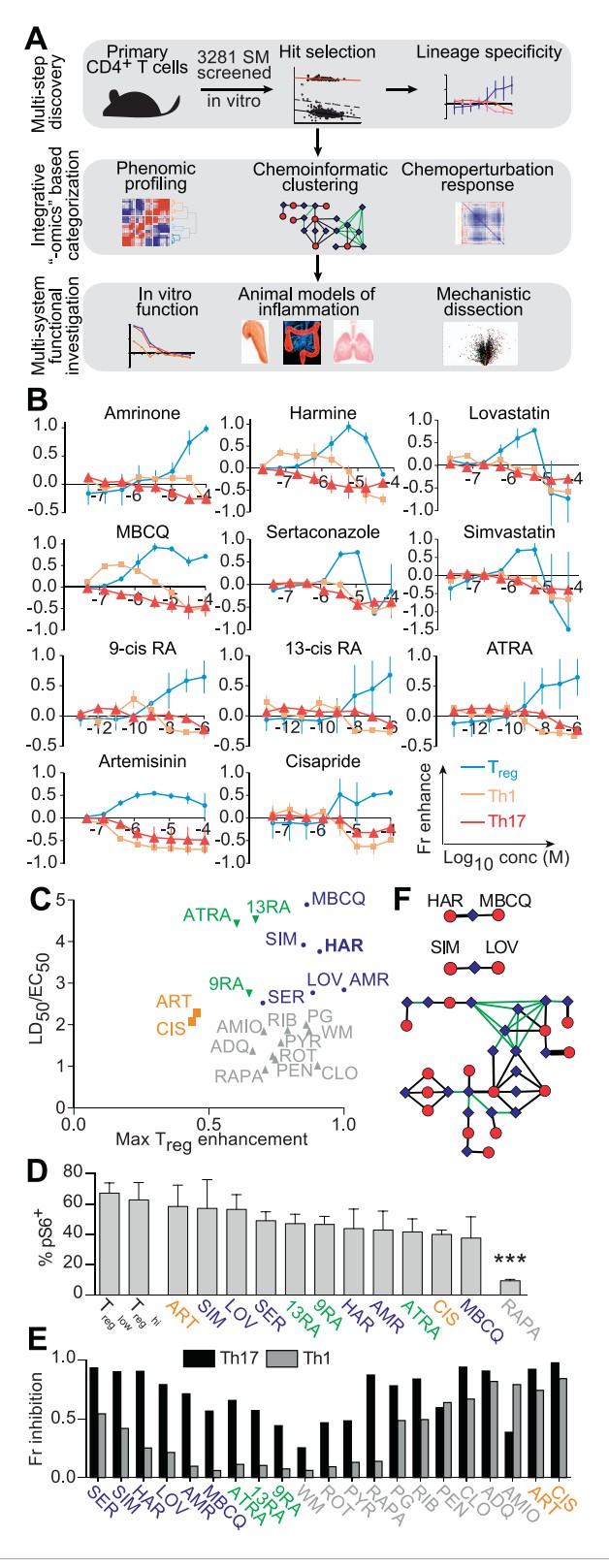

**Figure 1**. Chemical biology approach to identify novel $T_{reg}$ enhancers. All data representative of at least 2 independent experiments. (**A**) Overview of our approach, including key methods applied. (**B**) Dose-response curves showing fractional enhancement (Fr enhance) of compounds ($LD_{50}/EC_{50} > 2$) for $T_{reg}$ (blue), Th1 (orange) and Th17 (red) lineages. (**C**) Plot of $LD_{50}/EC_{50}$ vs maximal fractional $T_{reg}$ cell enhancement showing all 21 $T_{reg}$-specific

*Figure 1. continued on next page*

*Figure 1. Continued*

enhancers (9RA, 9-*cis* retinoic acid; 13RA, 13-*cis* retinoic acid; ADQ, amodiaquine; AMIO, amiodarone; AMR, amrinone; ATRA, all-*trans* retinoic acid; ART, artemisinin; CIS, cisapride; CLO, clotrimazole; HAR, harmine; LOV, lovastatin; MBCQ, 4-((3,4-methylenedioxybenzyl)amino)-6-chloroquinazoline;4-quinazolinamine); PEN, pentamidine; PG, proguanil; PYR, pyrvinium pamoate; RAPA, rapamycin; RIB, ribavirin; ROT, rotenone; SER, sertaconazole; SIM, simvastatin; WM, wortmannin; *Supplementary file 2*). Retinoic acids are in green; compounds with $LD_{50}/EC_{50} < 2$ are in gray and $LD_{50}/EC_{50} > 2$ are in blue. The orange cluster is described in the text. (**D**) Ability of selected compounds ($LD_{50}/EC_{50} > 2$ and controls) to inhibit mTOR activity, as measured by S6 phosphorylation (***$p < 0.001$, 1-way ANOVA with Dunnett correction). (**E**) Fractional (Fr) inhibitory activity of all 21 $T_{reg}$-specific enhancers on Th1 and Th17 cell differentiation. (**F**) SEA-predicted relationships between all 21 $T_{reg}$ enhancers. Black lines predict binding of compounds (red circles) to proteins (blue diamonds) with likelihood proportional to line width. Green lines denote connection via curated KEGG pathways. See also *Figure 1—figure supplements 1–8*.

The following figure supplements are available for figure 1:

**Figure supplement 1**. Titrating Th differentiation conditions.

**Figure supplement 2**. Culture cellularity affects $T_{reg}$ differentiation.

**Figure supplement 3**. Schematic of analytic and hit-calling pipeline.

**Figure supplement 4**. Effect of compounds ($LD_{50}/EC_{50} < 2$) on Th differentiation.

**Figure supplement 5**. Modeling analyses to calculate $EC_{50}$ values, indicated in parentheses, for all 21 $T_{reg}$-specific enhancers.

**Figure supplement 6**. Modeling analyses to calculate $LD_{50}$ values, indicated in parentheses, for all 21 $T_{reg}$-specific enhancers.

**Figure supplement 7**. Similarity clustering analysis of combined $T_{reg}$, Th1 and Th17 phenotypic data.

**Figure supplement 8**. Euclidean distance clustering analysis of gene expression data from cell lines treated with $T_{reg}$ enhancers first analyzed by principal component analysis.

Th17[low] conditions to quantitate their ability to enhance differentiation of pro-inflammatory lineages. Our results also consolidate and validate that the previously described $T_{reg}$ enhancers, including the statins, retinoic acids and artemisinin, specifically enhance $T_{reg}$ differentiation. Simultaneously testing multiple Th lineages and drug concentrations allows a more complete characterization of the effects of a compound on Th differentiation and furthers mechanistic conclusions. We identified 14 compounds hitherto unreported to specifically enhance differentiation of $T_{reg}$, but neither Th1 nor Th17, cells (*Figure 1B,C* and *Figure 1—figure supplement 4*).

## Novel $T_{reg}$ cell enhancers are mechanistically distinct from rapamycin

To demonstrate that these novel $T_{reg}$ cell enhancers work distinctly from known canonical pathways, we assessed their activity on mTOR activity. We selected this pathway for three primary reasons. Firstly, the role of mTOR inhibition on enhancing $T_{reg}$ cell differentiation has been well described. Secondly, rapamycin is a well-known mTOR inhibitor and enhancer of $T_{reg}$ cell differentiation, and thus provides a good positive control. Finally, mTOR activity is relatively easily assessed, for example by assessing the phosphorylation state of S6, which is phosphorylated in the course of mTOR activation. Thus, mTOR inhibition leads to fewer phospho-S6+ cells by flow cytometry. We characterized the effect of all 21 $T_{reg}$ cell enhancers on S6 phosphorylation in primary CD4+ T cells cultured under $T_{reg}$[low] conditions (*Figure 1D*). Only rapamycin, the positive control, significantly inhibited S6 phosphorylation (1-way ANOVA with Dunnett correction, threshold $p < 0.05$). Thus, all 14 novel $T_{reg}$ cell enhancers appear to work independently of mTOR and potentially point to undiscovered mechanisms.

## Prioritizing $T_{reg}$ cell enhancers for further investigation

We sought to identify compounds with minimal impact on cellularity, given its relationship with $T_{reg}$ cell differentiation. To this end, the $LD_{50}$ and $EC_{50}$ were determined as the doses at which 50% cytotoxicity and 50% $T_{reg}$ cell enhancement are observed, respectively (*Figure 1—figure supplements 5, 6*). Compounds were classified according to both the $LD_{50}/EC_{50}$ ratio (analogous to the therapeutic index) and the maximal enhancement of $T_{reg}$ cell differentiation, with ideal compounds performing maximally for both parameters (*Figure 1C*, blue circles). Many compounds, including rapamycin, exhibited significant cytotoxicity with an $LD_{50}/EC_{50}$ ratio near 1 (*Figure 1C*, gray triangles). All 21 $T_{reg}$ cell-specific enhancers were additionally tested under Th1$^{hi}$ and Th17$^{hi}$ conditions to accurately quantitate their capacity to inhibit differentiation into these pro-inflammatory lineages (*Figure 1E*). Th17 cell differentiation was typically more inhibited ($\geq$40%) than Th1, likely related to $T_{reg}$ cells and Th17 cells arising from a common progenitor (*Figure 1E*) (*Zhou et al., 2008*).

Unsupervised analysis of this phenotypic data revealed high similarity between artemisinin, cisapride and sertaconazole, including moderate enhancement of $T_{reg}$ cell differentiation and potent inhibition of both Th1 and Th17 cell differentiation, which may reflect effects on common pathways and direct future studies (*Figure 1C* and *Figure 1—figure supplement 7*). Importantly, these results prioritized MBCQ, harmine, and amrinone as novel enhancers of $T_{reg}$ cell differentiation with favorable phenotypic profiles (*Figure 1—figure supplement 7*).

In order to further explore potential mechanistic relationships between these $T_{reg}$ cell enhancers, we applied previously described chemoinformatic approaches. Similarity Ensemble Approach (SEA) is one such method that utilizes similarities in chemical structure between compounds to predict the likelihood that they could bind common protein targets that have been defined in previous efforts (*Keiser et al., 2007*). Applying SEA to our list of $T_{reg}$ cell enhancers generated several clusters of compounds predicted to bind the same protein (*Figure 1F*, black lines). Additionally, we recognized a need to account for relationships between these clusters, for example with compounds acting on separate proteins that act within the same pathway. These connections were identified using a manually curated list of KEGG pathways that excludes overly generic and largely populated pathways (e.g., Pathways in Cancer) that would report spurious relationships (*Figure 1F*, green lines) (*Goel et al., 2014*). These results suggested inter-relationships between most of our compounds with two outlier pairs, one comprising the statins and the other comprising harmine and MBCQ (*Figure 1F*).

The L1000 method had previously been used to generate gene expression data from three different cell lines treated with numerous compounds, including most of the $T_{reg}$ cell enhancers identified here (expression data in GSE5258) (*Lamb et al., 2006*). Principal component analysis was used to analyze changes in gene expression after treatment with $T_{reg}$ cell enhancers. These results indicated significant commonalities between most of the $T_{reg}$ cell enhancers identified here, with harmine and the retinoic acids generating the most distinct profiles (*Figure 1—figure supplement 8*). Together, our results prioritize harmine for its favorable phenotypic profile and likelihood of mechanistic novelty, given its distinct properties in our chemoinformatic analyses.

## Harmine-enhanced $T_{reg}$ cells suppress T cell proliferation in vitro

To further validate our interest in the physiologic relevance of harmine's ability to enhance $T_{reg}$ cell differentiation, we tested the functionality of $T_{reg}$ cells generated under $T_{reg}^{low}$ + harmine (henceforth abbreviated as $T_{reg}^{HAR}$) conditions extensively. First, we used an in vitro suppression assay, where $T_{reg}$ cells are co-cultured at increasing dilutions with CFSE-labeled responder CD4$^+$ T cells and their ability to suppress responder T cell proliferation upon anti-CD3/CD28 stimulation is assessed. Naïve CD4$^+$ T cells from *Foxp3*$^{IRES-GFP}$ mice were cultured under either $T_{reg}^{HAR}$ or $T_{reg}^{hi}$ conditions and the resulting GFP$^+$ $T_{reg}$ cells were sorted by FACS. Sorted $T_{reg}$ cells generated using either $T_{reg}^{hi}$ or $T_{reg}^{HAR}$ conditions equivalently suppressed the proliferation of co-cultured responder CD4$^+$ T cells in vitro at each dilution, indicating equal efficacy between both populations of $T_{reg}$ cells (*Figure 2A*, red and blue lines). Both populations worked better than sorted t$T_{reg}$ cells, likely because they are pre-activated (*Figure 2A*, orange line).

## Harmine-enhanced $T_{reg}$ cells attenuate disease in three experimental models of inflammation

We also compared the ability of $T_{reg}^{hi}$- and $T_{reg}^{HAR}$-$T_{reg}$ cells to inhibit inflammation in vivo. For this purpose, we selected three experimental models in which inflammation is mediated by T cells and can

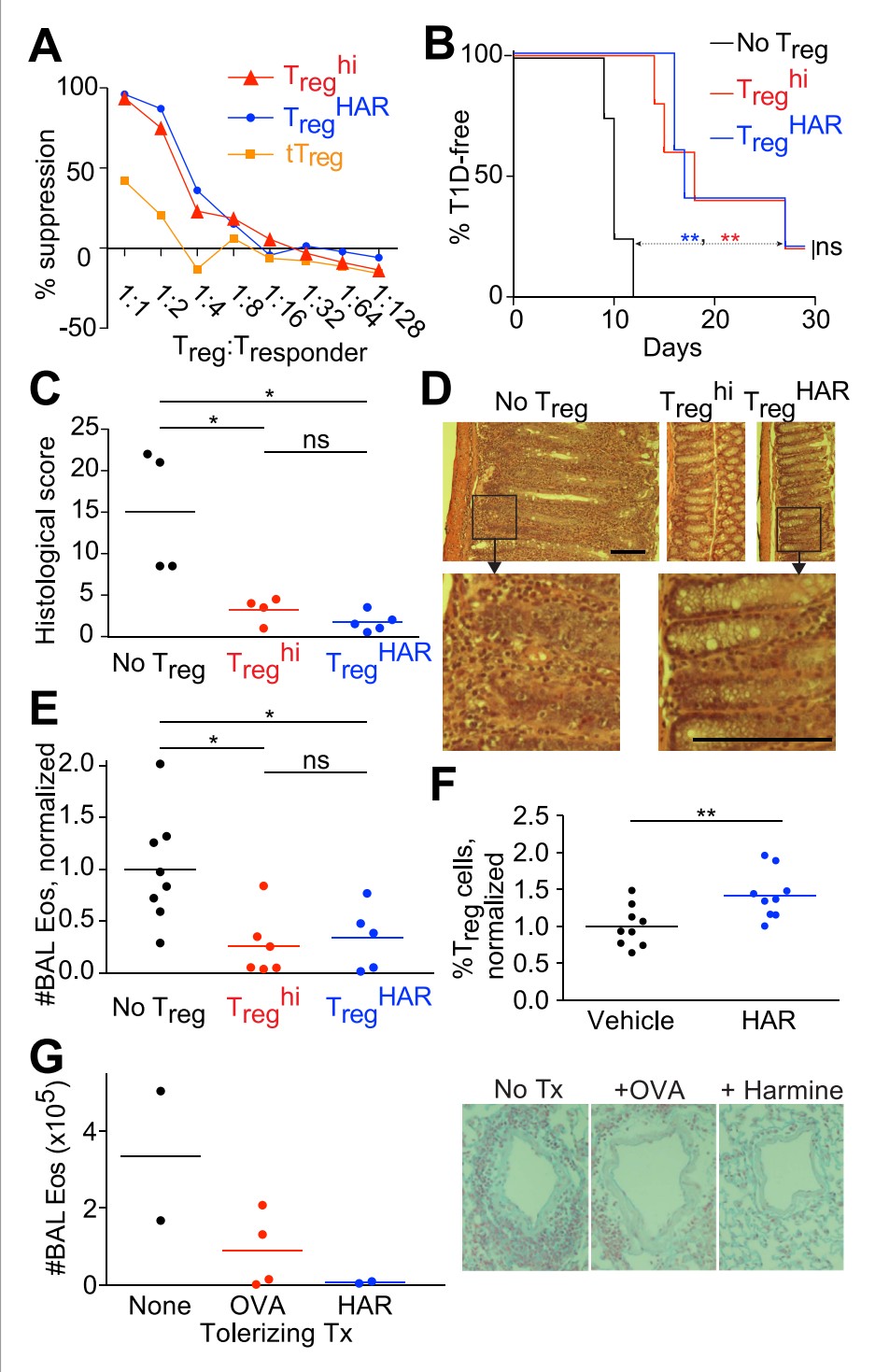

**Figure 2**. Harmine-enhanced $T_{reg}$ cells and harmine attenuate inflammation. (**A–E**) Experiments comparing the suppressive activity of $T_{reg}$ cells generated under either $T_{reg}^{HAR}$ (blue) or $T_{reg}^{hi}$ (red) conditions; conditions without $T_{reg}$ cells are shown in black. All data representative of at least 2 independent experiments; in vivo experiments used at least 3 mice per cohort. (**A**) In vitro suppression assay, with unstimulated $tT_{reg}$ cells in orange. (**B**) $T_{reg}^{hi}$-$T_{reg}$ cells and $T_{reg}^{HAR}$-$T_{reg}$ cells similarly delay onset of diabetes in the NOD-*BDC2.5* model of type 1 diabetes. (**C**) Comparable inhibition of inflammation by $T_{reg}^{hi}$-$T_{reg}$ cells and $T_{reg}^{HAR}$-$T_{reg}$ cells in the CD45RB$^{hi}$ transfer model of colitis. Representative images are shown in (**D**). Bars represent 100 µm. (**E**) Similar inhibition of airway inflammation by $T_{reg}^{hi}$-$T_{reg}$ cells and $T_{reg}^{HAR}$-$T_{reg}$ cells, as measured by number of eosinophils (Eos) in bronchoalveolar lavage (BAL)
*Figure 2. continued on next page*

*Figure 2. Continued*

fluid, in a model of asthma. (**F**) Comparison of T$_{reg}$ cells (as a percentage of total CD4$^+$ T cells) in the thoracic lymph nodes of mice treated with intranasal harmine, vs mock treatment. (**G**) Intranasal administration of harmine prior to immunization inhibits recall airway inflammation in the asthma model. Right, representative images of inflammation around airway vessels. \*\*p < 0.01, \*p < 0.05, ns, not significant, Mantel-Cox test (**B**, **C**) and Student's t-test (**E**, **F**). See also *Figure 2—figure supplements 1–2*.

The following figure supplements are available for figure 2:

**Figure supplement 1**. Effects of harmine treatment in vivo on T cell populations.

**Figure supplement 2**. Effects of harmine treatment in vivo on antigen-presenting cell populations.

be suppressed by T$_{reg}$ cells. These models were also selected to represent different genetic backgrounds, inflammation in different sites and different T$_{reg}$ cell antigen specificities.

In a model of T1D induced by transfer of NOD-*BDC2.5*$^+$ CD4$^+$ T cells into NOD-*scid* recipients, diabetes developed rapidly approximately 10 days later without any intervention (*Figure 2B*, black line). When antigen-specific T$_{reg}$ cells generated from NOD-*BDC2.5.Foxp3*$^{IRES-GFP}$ mice under either T$_{reg}$$^{HAR}$ or T$_{reg}$$^{hi}$ conditions were co-transferred, a consistent and indistinguishable delay of onset of diabetes was observed, with median time of diabetes onset being delayed by at least 7 days (*Figure 2B*, blue and red lines) (*Herman et al., 2004*; *Tarbell et al., 2004*).

Using a well-described T cell-dependent model of colitis, transfer of C57Bl/6 CD4$^+$CD45RB$^{hi}$ T cells into C57Bl/6-*Rag1*$^{–/–}$ hosts led to intestinal inflammation 8 weeks later (*Figure 2C,D*, no T$_{reg}$) (*Powrie et al., 1993*). This mucosal inflammation was significantly attenuated when antigen-naïve T$_{reg}$ cells generated from C57Bl/6-*Foxp3*$^{IRES-GFP}$ mice under T$_{reg}$$^{hi}$ conditions were transferred, as determined by histological scoring of intestinal sections by blinded observers (*Figure 2C,D*) (*Smith et al., 2013*). Transfer of T$_{reg}$ cells generated under T$_{reg}$$^{HAR}$ conditions attenuated intestinal inflammation to a level indistinguishable from that achieved by transfer of T$_{reg}$$^{hi}$-T$_{reg}$ cells (*Figure 2C,D*).

Finally, the functionality of T$_{reg}$$^{HAR}$-T$_{reg}$ cells was tested in a model of airway inflammation. Here, C57Bl/6 mice are sensitized against ovalbumin and subsequent challenge with intratracheally administered ovalbumin leads to airway inflammation (*Figure 2E*) (*Grainger et al., 2010*). This inflammation was attenuated when antigen-naïve T$_{reg}$ cells generated from C57Bl/6-*Foxp3*$^{IRES-GFP}$ mice under T$_{reg}$$^{hi}$ conditions were transferred prior to the intratracheal challenge (*Figure 2E*). Importantly, transfer of T$_{reg}$ cells generated under T$_{reg}$$^{HAR}$ conditions led to a comparable suppression of inflammation (*Figure 2E*).

## Harmine promotes T$_{reg}$ cell differentiation in vivo and attenuates airway inflammation

The observation that harmine promotes the differentiation of T$_{reg}$ cells, at least in vitro, that appear fully functional raises the interesting hypothesis that treatment with harmine itself could attenuate inflammation in vivo. Rapid first pass metabolism (<2 hr) consistent with prior reports confounded the interpretation of systemic delivery experiments (*Callaway et al., 1999*). Reasoning that application of harmine to mucosal surfaces might allow for relatively prolonged local presence, we treated mice with harmine intranasally for 5 days and examined the effect on T$_{reg}$ cells in the draining lymph nodes. Compared to mice treated only with vehicle (water), mice treated with harmine exhibited a statistically significant increase (~20%) in the frequency of T$_{reg}$ cells in the draining thoracic lymph nodes; increases in absolute numbers of effector T cell subsets did not reach statistical significance (*Figure 2F* and *Figure 2—figure supplement 1*). Analyses of dendritic cell populations did not show any effect of treatment with harmine on expression of T$_{reg}$-relevant costimulatory molecules, consistent with the notion that harmine predominantly acts directly on CD4$^+$ T cells to affect T$_{reg}$/Th17 differentiation in this model (*Figure 2—figure supplement 2*). To determine if this pro-T$_{reg}$ effect might impact inflammation, we adapted the model of airway inflammation described above, where sensitivity to ovalbumin is induced by immunization. Intranasal administration of ovalbumin 5–7 days prior to immunization attenuated the airway inflammation induced by subsequent intratracheal challenge (*Figure 2G*). This finding supports the notion that exogenous signals can modulate the inflammatory response mounted

at the time of immunization. Strikingly, intranasal administration of only harmine during this window inhibited airway inflammation at least as potently as tolerization with ovalbumin (*Figure 2G*).

## Harmine reciprocally regulates T$_{reg}$ and Th17 cell differentiation through a novel mechanism

Our results demonstrate that harmine is a novel, potent, and specific enhancer of T$_{reg}$ cell differentiation with physiologically relevant effects (*Figures 1B, 3A*). In addition to its pro-T$_{reg}$ effect, harmine strongly inhibits Th17 cell differentiation (*Figure 3A*). Notably, even in pro-inflammatory Th17$^{low}$ or Th17$^{hi}$ conditions, harmine modestly promotes the paradoxical differentiation of T$_{reg}$ cells approximately twofold (*Figure 3A*). At the doses used, harmine does not significantly affect culture cellularity, unlike ATRA and rapamycin (*Figure 3B*). This observation is further substantiated by CFSE studies of cellular proliferation, which show that harmine causes a modest 24% reduction in proliferating cells at day 3 that falls to undetectable levels by day 4, less than half the reduction caused by rapamycin (*Figure 3—figure supplement 1*). Accordingly, harmine enhances absolute numbers of T$_{reg}$ cells to levels approaching T$_{reg}$$^{hi}$ conditions and decreases absolute numbers of Th17 cells (*Figure 3B* and *Figure 3—figure supplement 2*). Importantly, similar effects are observed using human CD4$^+$ T cells, with addition of harmine potently enhancing both percentage and absolute numbers of T$_{reg}$ cells beyond even T$_{reg}$$^{hi}$ conditions (*Figure 3C*). These findings underscore the physiologic relevance of harmine-related pathways to human Th differentiation.

In the context of neutral Th0 conditions, addition of harmine does not skew Th differentiation towards any lineage, demonstrating that harmine's pro-T$_{reg}$ effect requires exogenous TGF-β1 (*Figure 3D*). Thus, harmine does not substitute for TGF-β1 but rather acts in conjunction with TGF-β1. In order to better characterize how harmine acts to enhance T$_{reg}$ cell differentiation, CD4$^+$ T cells were cultured in T$_{reg}$$^{low}$ conditions and harmine was added 1, 2 or 3 days later. If the molecular targets of harmine were only expressed early in Th differentiation, then addition of harmine at later time points would have no effect on T$_{reg}$ cell differentiation. Our results show that adding harmine as late as day 3 (out of 4) of culture still significantly enhances T$_{reg}$ cell differentiation (*Figure 3E*, top panel). The converse experiments, where culture is initiated in T$_{reg}$$^{HAR}$ conditions and harmine removed 4 hr, 1 day or 2 days later, showed complementary results. The earlier harmine was removed, the less T$_{reg}$ cell differentiation was enhanced, although enhancement could be detected with as little as 4 hr of exposure to harmine (*Figure 3E*, bottom panel). These results not only reinforce the conclusion that the targets of harmine that impact Th differentiation are present throughout the process of differentiation, but also demonstrate that harmine does not impart long-lasting epigenetic signals. If that were the case, maximal T$_{reg}$ cell enhancement would be observed even if harmine were removed after a short time. Corresponding reciprocal results were obtained when harmine was either added or removed at different times in the context of Th17 cell differentiation (*Figure 3—figure supplement 3*).

To determine the effect of harmine on gene expression in T$_{reg}$ cells, RNA was isolated from FACS-sorted T$_{reg}$ cells, generated under either T$_{reg}$$^{HAR}$ or T$_{reg}$$^{hi}$ conditions, and analyzed by Illumina microarray (data in GSE67961). These results revealed significant similarity between the expression profiles of T$_{reg}$$^{hi}$- and T$_{reg}$$^{HAR}$-T$_{reg}$ cells (Pearson correlation coefficient = 0.95, *Figure 3—figure supplement 4*). As might be expected from this result, T$_{reg}$$^{HAR}$-T$_{reg}$ cells showed concordant regulation of previously described canonical T$_{reg}$ cell signature genes (*Figure 3F*) (*Feuerer et al., 2010*). We found no evidence of significant similarity to T$_{reg}$ cells specialized to suppress Th1, Th2 or Th17 cells (CXCR3$^+$, IRF4$^+$ and GFP-FOXP3-fusion T$_{reg}$ cells respectively, *Figure 3—figure supplement 5*) (*Joller et al., 2014*). However, compared to T$_{reg}$$^{hi}$-T$_{reg}$ cells, T$_{reg}$$^{HAR}$-T$_{reg}$ cells showed a bias suggesting increased activation (*Figure 3G*) (*Joller et al., 2014*). In addition, flow cytometry studies showed that, after gating on FOXP3$^+$ T$_{reg}$ cells, T$_{reg}$$^{HAR}$-T$_{reg}$ cells express FOXP3 at levels at least as high as, if not higher than, those of T$_{reg}$$^{hi}$-T$_{reg}$ cells (*Figure 3H*). Together, these results suggest that harmine promotes the differentiation of T$_{reg}$ cells that are of similar to superior function as compared to those driven by high levels of TGF-β1 alone.

## Harmine does not impact canonical pathways of T$_{reg}$ cell differentiation

We next examined harmine's effects on 3 canonical pathways of T$_{reg}$ cell differentiation, namely FOXP3 expression, mTOR activity and TGF-β1/SMAD signaling. To determine if harmine promotes earlier expression of FOXP3, we analyzed FOXP3 expression by flow cytometry at daily intervals

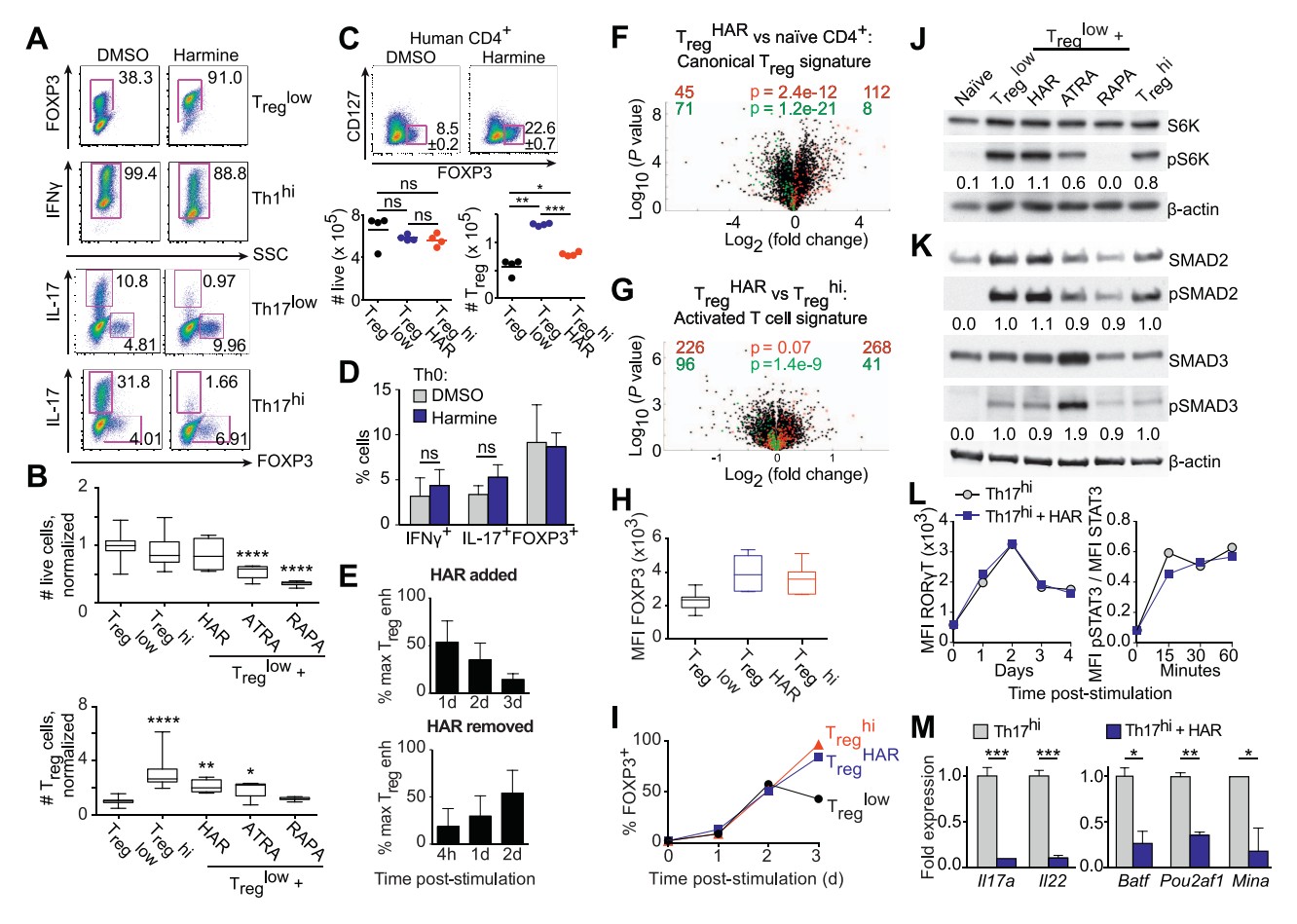

**Figure 3.** Harmine's effects on canonical $T_{reg}$/Th17 pathways. All data representative of at least 2 independent experiments. (**A**) Effect of harmine on murine $T_{reg}$, Th1 and Th17 differentiation. (**B**) Effect of harmine on absolute numbers of total live and $T_{reg}$ cells. (**C**) Effect of harmine on human $T_{reg}$ differentiation. (**D**) Effect of harmine on Th differentiation under Th0 conditions as shown by percentage of cells with indicated markers. (**E**) Harmine's pro-$T_{reg}$ effect when added or removed at different times after $T_{reg}^{low}$ stimulation as shown by percentage of maximal $T_{reg}$ enhancement (% max $T_{reg}$ enh). (**F** and **G**) Volcano plots comparing p value vs fold change in gene expression in naïve CD4+ T cells, $T_{reg}^{hi}$-$T_{reg}$ cells and $T_{reg}^{HAR}$-$T_{reg}$ cells as indicated. Previously reported signature genes for $T_{reg}$ cells (**F**) and activated T cells (**G**) are highlighted in red (upregulated) and green (downregulated). Numbers on the right and left reflect genes that are up- and down-regulated in the indicated comparison respectively with $\chi^2$ test p values in the middle. (**H**) Median fluorescence intensity (MFI) of FOXP3 in $T_{reg}$ cells generated under indicated conditions. (**I**) Time-course analysis of FOXP3 expression in cells cultured under indicated conditions. (**J** and **K**) Western blot analyses showing effect of $T_{reg}$ enhancers on S6 kinase, SMAD2 and SMAD3 phosphorylation when added under $T_{reg}^{low}$ conditions. Numbers denote fractional phosphorylation relative to $T_{reg}^{low}$ conditions. (**L**) Effect of harmine (blue) on RORγT expression and STAT3 phosphorylation in Th17$^{hi}$ conditions (gray). (**M**) qPCR analyses showing effects of harmine on key Th17 genes at 4 days (left) and 2 hours (right) after stimulation. Gray and blue bars represent Th17$^{hi}$ and Th17$^{hi}$ + harmine conditions, respectively. *p < 0.05, **p < 0.01, ***p < 0.001, ****p < 0.0001, ns, not significant, Student's t-test with Holm-Sidak correction (**B**, **C**, **D**, **M**). See also *Figure 3—figure supplements 1–5*.

The following figure supplements are available for figure 3:

**Figure supplement 1**. Effect of harmine on cellular proliferation.

**Figure supplement 2**. Effect of harmine on absolute numbers of Th17 cells.

**Figure supplement 3**. Effect of addition or removal of harmine at different times on Th17 differentiation.

**Figure supplement 4**. Correlation between gene expression in $T_{reg}^{hi}$-$T_{reg}$ cells and $T_{reg}^{HAR}$-$T_{reg}$ cells, relative to naïve CD4+ T cells.

**Figure supplement 5**. Qualitative analyses of genomewide expression in $T_{reg}^{HAR}$-$T_{reg}$ cells.

during T_reg cell differentiation. The kinetics of FOXP3 expression between T_reg^low, T_reg^hi and T_reg^HAR conditions were indistinguishable between days 0–2 (*Figure 3I*). At day 3, the percentage of FOXP3+ cells consistently decreased in T_reg^low conditions, while increasing identically in T_reg^hi and T_reg^HAR conditions (*Figure 3I*). These results argue against the notion that harmine enhances T_reg cell differentiation by driving earlier expression of FOXP3. They also suggest that high levels of TGF-β1 do not accelerate the early kinetics of FOXP3 expression, and the enhanced T_reg cell differentiation seen in T_reg^hi conditions may reflect either stabilization of the FOXP3+ state and/or higher TGF-β1 levels available at later timepoints to continue driving T_reg cell differentiation.

To complement our flow cytometric studies of mTOR activity, we measured phosphorylation of another protein, S6-kinase, by Western blot. Again, only rapamycin inhibited S6-kinase phosphorylation, confirming that harmine does not inhibit mTOR activity (*Figure 3J*).

Finally, we measured phosphorylation of SMAD2 and SMAD3 to determine if harmine enhances TGF-β1 signaling through these canonical molecules. Phosphorylation of both SMAD2 and SMAD3 was increased upon stimulation under T_reg^low conditions compared to naïve CD4+ T cells, consistent with engagement of TGF-β1 signals (*Figure 3K*). As previously reported, addition of ATRA leads to increased phosphorylation of SMAD3, but not SMAD2 (*Figure 3K*) (*Xiao et al., 2008*). Importantly, there was no further increase in SMAD2/3 phosphorylation upon further addition of harmine, rapamycin, or TGF-β1 (*Figure 3K*). Taken together, these data indicate that harmine does not enhance T_reg cell differentiation by amplifying signaling through these canonical pathways. Notably, our finding that T_reg^hi conditions do not enhance SMAD2/3 phosphorylation beyond T_reg^low conditions further demonstrates that TGF-β1 itself engages pertinent and quantitative SMAD2/3-independent signals outside of tT_reg cells (*Figure 3K*).

## Harmine does not impact canonical pathways of Th17 cell differentiation

Th17 cell differentiation centrally involves IL-6 signaling through STAT3, which in turn leads to expression of RORγT, the hallmark Th17 transcription factor. Kinetic analyses showed increased STAT3 phosphorylation and RORγT expression upon activation in Th17^hi conditions (*Figure 3L*). No difference was observed when harmine was added, indicating that neither signaling event is affected by harmine (*Figure 3L*). We verified by qPCR that harmine inhibits expression not only of the effector molecules *Il17a* and *Il22*, but also of several key regulators of the Th17 pathway, including *Batf*, *Pou2af1* and *Mina* (*Figure 3M*) (*Schraml et al., 2009*; *Yosef et al., 2013*). These results suggest that harmine works on novel target(s) to modulate T_reg and Th17 cell differentiation and highlight a druggable point between STAT3/RORγT signaling and other Th17 transcription factors.

## Generating a genetic signature of harmine-enhanced T_reg cells suggests relevance to IBD

To gain insight into harmine-regulated genes and pathways, we compared the expression profiles of T_reg^HAR-T_reg cells and T_reg^hi-T_reg cells. Since harmine also inhibits Th17 cell differentiation, we focused on a 111-gene signature that was concordantly up/down-regulated in T_reg^HAR-T_reg cells vs T_reg^hi-T_reg cells, as well as in human T_reg cells vs Th17 cells (*Figure 4A*) (*Zhang et al., 2013*). To assess if these effects might be relevant to human disease, we evaluated the overlap between these 111 genes (and the 16 transcription factors whose binding sites were overrepresented therein) with the 1437 genes that lie within IBD-associated loci as previously reported (*Jostins et al., 2012*; *Okada et al., 2014*). The overlap of 16 genes represented a significant (p < 0.01) enrichment of 1.76-fold, suggesting that harmine impacts T_reg cell-relevant genes implicated in IBD (*Figure 4B*). Of these, we were particularly interested in BACH2, a transcription factor linked to T_reg cell development and inflammation, as well as to pediatric IBD (*Christodoulou et al., 2013*; *Roychoudhuri et al., 2013*). BACH2-deficient mice exhibit a progressive wasting disease with autoantibody formation and inflammation in the lung and gut leading to decreased survival due, at least in part, to defective T_reg cell development and function (*Roychoudhuri et al., 2013*). Independent qPCR experiments confirmed differential expression of 5 of the 6 T_reg^HAR signature genes with predicted BACH2 binding sites, supporting the notion that harmine enhances T_reg cell differentiation at least in part by modulating BACH2 signaling (*Figure 4C*). During polarization of naïve CD4+ T cells under pro-T_reg conditions, BACH2 stabilizes T_reg differentiation by suppressing transcriptional programmes associated with other Th lineages; a corresponding BACH2-dependent signature has been identified (*Roychoudhuri et al., 2013*). Intriguingly, a significant number of these BACH2-regulated genes are inversely regulated by harmine, suggesting that harmine may help reverse BACH2-axis defects, for example in IBD (*Figure 3—figure supplement 5*).

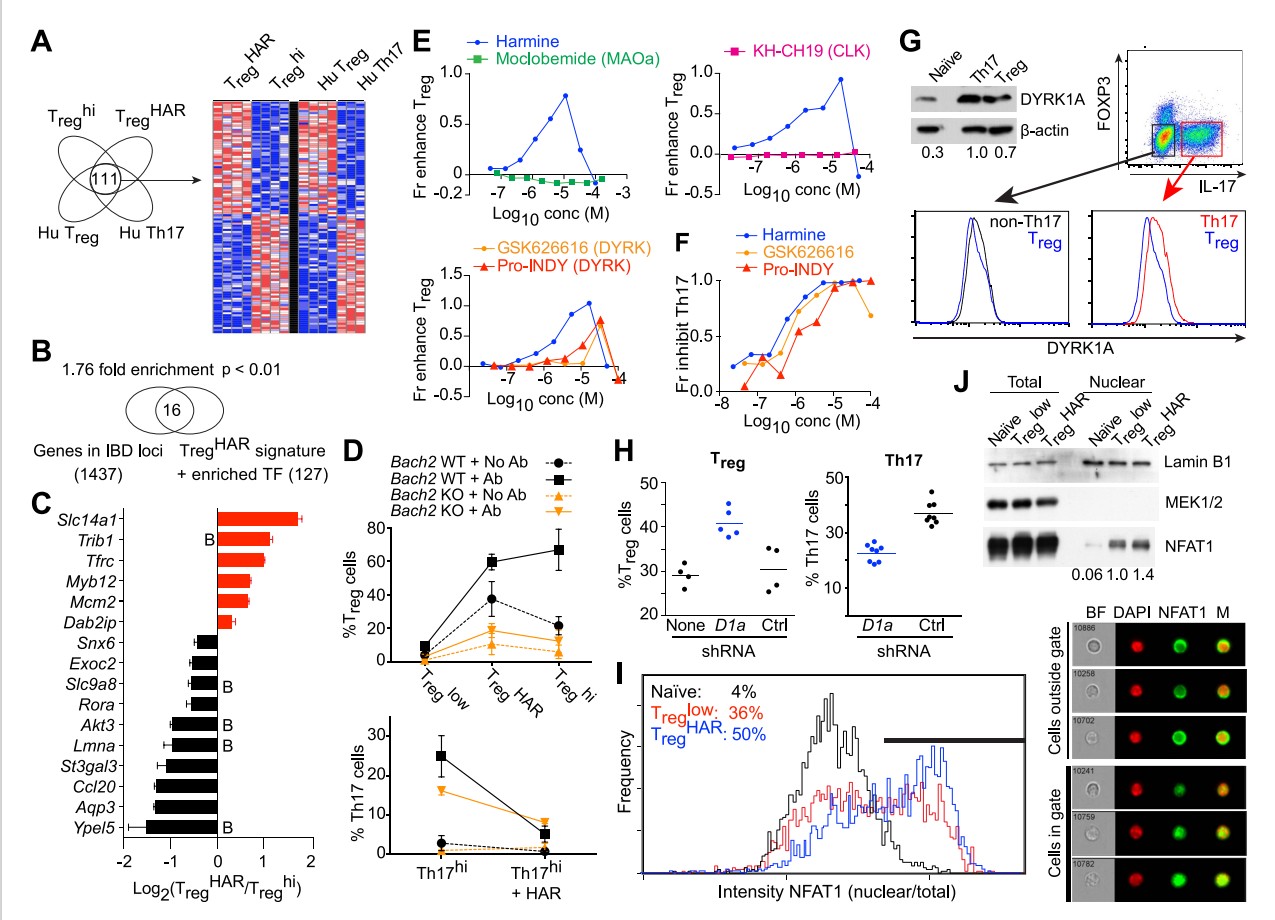

**Figure 4**. Mechanistic dissection of harmine. (**A**) Comparison of expression profiles suggesting a harmine-relevant $T_{reg}$ signature. The black bar separates 2 independently row-normalized experiments. (**B**) Harmine signature genes are enriched for genes in IBD loci. (**C**) Validation of top signature genes by qPCR (all p < 0.05, Student's t-test), including genes with BACH2 binding sites (B). (**D**) Effect on harmine on $T_{reg}$/Th17 differentiation in BACH2-deficient (orange) vs -sufficient (black) cells. Conditions in the presence and absence of neutralizing antibodies are indicated by solid and dotted lines, respectively. (**E**) Effect of compounds that inhibit different harmine targets (indicated in parentheses) on $T_{reg}$ cell differentiation. (**F**) DYRK inhibitors suppress Th17 cell differentiation. (**G**) Increased levels of DYRK1A in Th17 vs $T_{reg}$ cells. Upper left, Western blot analyses of sorted $T_{reg}$/Th17 cells with relative expression enumerated below. Histograms show FACS analyses of DYRK1a in $T_{reg}$ cells (blue) compared to either Th17 (red) or non-Th17 (black) cells. (**H**) Knock-down of *Dyrk1a* (*D1a*) enhances $T_{reg}$ (left) and inhibits Th17 (right) cell differentiation. Cells treated with non-targeting shRNA (Ctrl) and no shRNA (None) are shown for comparison. (**I**) Amnis analyses showing nuclear-overlapping NFAT1 signal before (black) and after stimulation in $T_{reg}^{low}$ conditions with (blue) or without (red) harmine. Representative images (right) illustrate cytoplasmic and nuclear NFAT1 localization in cells outside and within the gate, respectively. (**J**) Western blot analyses quantitating nuclear fraction of NFAT1 in cells stimulated in $T_{reg}^{low}$ ± harmine conditions. All data in **D**–**J** representative of at least 2 independent experiments. See also *Figure 4—figure supplements 1–3*.

The following figure supplements are available for figure 4:

**Figure supplement 1**. Effect of harmine on NFAT1 nuclear localization with time.

**Figure supplement 2**. Comparing effects of DYRK1A deficiency to harmine treatment.

**Figure supplement 3**. Secondary analyses of effects of DYRK1A deficiency compared to harmine treatment.

To further dissect this link, we examined the effect of harmine on $T_{reg}$/Th17 differentiation in BACH2-deficient T cells. Mixed bone marrow chimeras allowed us to simultaneously examine the effect of harmine on wildtype and BACH2-deficient T cells. Our studies reproduced the previously reported cell-extrinsic defect in $T_{reg}$ (and an even more impressive defect in Th17) differentiation which is due in part to the dysregulated production of cytokines like IFNγ by BACH2-deficient cells; this can be

attenuated by adding neutralizing antibodies against IFNγ and IL-4 (*Figure 4D*, dotted vs solid lines) (*Roychoudhuri et al., 2013*). Importantly, harmine enhances T$_{reg}$ differentiation and inhibits Th17 differentiation in both wildtype and BACH2-deficient cells, indicating that harmine engages BACH2-independent programs to regulate T$_{reg}$/Th17 differentiation (*Figure 4D*). Interestingly, harmine does not regulate the differentiation of BACH2-deficient cells to the same level as wildtype cells, consistent with the notion that harmine also works, at least in part, through BACH2-dependent mechanisms (*Figure 4D*).

## Harmine enhances T$_{reg}$ cell differentiation by inhibiting DYRK activity

Harmine inhibits the activity of several targets including monoamine oxidase A (MAOA), CDC-like kinases (CLKs) and dual-specificity tyrosine-phosphorylation regulated kinases (DYRKs) (*Bain et al., 2007*; *Aranda et al., 2011*). In order to identify those relevant to T$_{reg}$ cell differentiation, we tested compounds that differentially inhibit each of these targets. Again, each compound was tested at multiple doses to capture any effect. Inhibition of either MAOA or CLKs using moclobemide and KH-CH19, respectively, did not enhance T$_{reg}$ cell differentiation at any dose (*Figure 4E*) (*Fedorov et al., 2011*). However, two other DYRK inhibitors, GSK-626616 and pro-INDY, similarly enhanced T$_{reg}$ and inhibited Th17 cell differentiation (*Figure 4E,F*) (*Ogawa et al., 2010*; *Wippich et al., 2013*). The physiological relevance of this finding is supported by FACS and Western blot studies that reproducibly showed higher levels of DYRK1A in Th17 than T$_{reg}$ cells (*Figure 4G*). This difference is specific; non-Th17 cells generated in the context of pro-Th17 conditions do not exhibit elevated levels of DYRK1A (*Figure 4G*). Furthermore, knock-down of *Dyrk1a* in primary CD4$^+$ T cells resulted in increased differentiation of T$_{reg}$ and decreased differentiation of Th17 cells (*Figure 4H*). Together, these data point to a central role at least for DYRK1A in regulating T$_{reg}$ and Th17 cell differentiation.

DYRKs phosphorylate several proteins (*Aranda et al., 2011*). Notable amongst these in the context of T cell biology are NFAT proteins, whose phosphorylation by DYRKs leads to their nuclear exclusion (*Gwack et al., 2006*). Thus, inhibition of DYRKs would be predicted to lead to increased levels of NFAT in the nucleus. To assess this hypothesis, we performed studies using Amnis technology, which combines flow cytometry and high-resolution microscopy to allow precise quantitation of intracellular localization of individual proteins. Naïve CD4$^+$ T cells largely retain NFAT1 in the cytoplasm (*Figure 4I*, black line). Upon stimulation in T$_{reg}$$^{low}$ conditions, nuclear translocation of NFAT1 is observed with an accompanying right shift of the nuclear/cytoplasmic ratio (*Figure 4I*, red line). This nuclear translocation of NFAT1 is reproducibly enhanced approximately 40% with the addition of harmine (*Figure 4I*, blue line). In support of these results, we independently fractionated nuclei from cells treated with each of these conditions and performed Western blot analyses. These also showed increased NFAT1 in the nuclear compartment upon stimulation in T$_{reg}$$^{low}$ conditions, with a similar (40%) additional increase when harmine was added (*Figure 4J*). Thus, harmine enhances nuclear accumulation of NFAT1 at early time points up to 2 hr after stimulation (*Figure 4—figure supplement 1*).

To gain further insight into the pathways engaged by harmine and DYRK1A, we activated primary CD4$^+$ T cells under Th0 conditions followed either by transduction with *Dyrk1a*-specific shRNA or control shRNA followed by harmine treatment. RNAseq studies found significant similarities in the expression profiles subsequent to either *Dyrk1a* knockdown or harmine treatment (Pearson correlation coefficient = 0.65) and genes that were up- or down-regulated as a result of *Dyrk1a* knockdown were predominantly concordantly regulated by harmine (*Figure 4—figure supplement 2*, data in GSE67961). Neither *Dyrk1a* knockdown nor harmine treatment significantly regulated genes associated with either the canonical T$_{reg}$ signature or our 111-gene T$_{reg}$$^{HAR}$-T$_{reg}$ signature (*Figure 4—figure supplement 3*). This is likely related to our observation that harmine does not promote T$_{reg}$ differentiation in the absence of TGFβ; the differences observed here may be upstream of more T$_{reg}$-associated expression changes. A relatively small number of genes (150) were differentially regulated between *Dyrk1a* knockdown and harmine treatment, which may in part reflect ancillary mechanisms engaged by harmine to regulate T$_{reg}$/Th17 differentiation. Overall, our data are consistent with the notion that DYRK1A inhibition represents a major mechanism by which harmine regulates T$_{reg}$/Th17 differentiation.

## Discussion

T$_{reg}$ cells are an important regulator of immune homeostasis and an attractive therapeutic target because of their role in human inflammatory diseases such as IBD and T1D. Nevertheless, there remains

a lack of drugs as well as druggable genes and pathways that specifically modulate Th differentiation. Although there is significant interest in manipulating $T_{reg}$ cells to treat IBD and T1D, the most mature efforts are found in the setting of organ transplantation where there is still significant room for improvement, ideal $T_{reg}$ cell subpopulation properties are still unclear and rapamycin is the most cutting-edge compound being used (*Desreumaux et al., 2012*; *Long et al., 2012*; *Edozie et al., 2014*).

To address this issue, we report a systematic, high-throughput pipeline to investigate the effects of small molecules on Th cell differentiation. These efforts enabled us to build a comprehensive profile of how compounds affect T cell viability and differentiation into both pro- and anti-inflammatory Th subsets. Drug selection could be guided by such information; for example, an ideal anti-inflammatory drug would not enhance, and would preferably inhibit, Th differentiation into pro-inflammatory lineages. Indeed, the inclusion of many FDA-approved drugs in our studies illustrates the potential of this approach to be applied to drug repurposing efforts. In this regard, our results reinforce interest in clinically used hypolipidemic statins, including lovastatin and simvastatin, as pro-$T_{reg}$ cell and anti-Th17 compounds and suggest that their targets, whose geranylgeranylation are inhibited, are of fundamental and clinical interest (*Kagami et al., 2009*; *Kim et al., 2010*; *Zhang et al., 2008*). Clinical studies suggest that statins may be useful in rheumatoid arthritis, with somewhat more mixed results in systemic lupus erythematosus and multiple sclerosis (*Ulivieri and Baldari, 2014*).

Our cytotoxicity data identify significant toxicity with many known $T_{reg}$ cell enhancers, including rapamycin, supporting the value of a continued search. Furthermore, our computational approaches suggest a framework to bin compounds into mechanistic classes, which holds particular relevance to future efforts to use polypharmacy to modulate the immune response by suggesting combinations that might target the maximal breadth of inflammatory pathways.

These studies demonstrate the novel and simultaneous application of three key principles, namely unbiased chemical biology, maximally physiologic experimental system and selection of a phenotypic readout. The advantages of the first two have already been alluded to—studying more compounds intuitively increases the likelihood of discovering novel biology, assuming an accompanying increase in complexity of chemical structures tested, and our use of primary CD4$^+$ T cells, as opposed to a cell line, maximizes the likelihood of our findings being physiologically relevant. Importantly, using a phenotypic primary endpoint significantly extends the scope of previous chemical biology efforts which had largely centered around finding compounds that bind known key regulators of Th differentiation, such as RORγT (*Huh et al., 2011*; *Xiao et al., 2014*). Such studies hold therapeutic promise and have highlighted the utility of using larger chemical libraries. However, the nature of the question fundamentally limits the potential mechanistic insight to targets of the pre-identified key regulator. In contrast, our use of a phenotypic endpoint is designed to capture any compound that affects $T_{reg}$ cell differentiation regardless of mechanism. In proof of this concept, we now identify 14 compounds as novel and specific enhancers of $T_{reg}$ cell differentiation, the largest single addition to the $T_{reg}$ cell biologist's chemical toolkit. We fully anticipate that subsequent studies will elucidate these compounds' mechanisms of action, leading us to a fuller understanding of the pathways that regulate $T_{reg}$ cell differentiation. Already, some interesting themes can be observed in this set of $T_{reg}$ cell enhancers. Aside from the retinoic acids and statins, a significant number of them are antimicrobial agents, in particular antifungal agents (including sertaconazole, clotrimazole and pentamidine) and antimalarials (artemisinin, amodiaquine and proguanil). This observation suggests how such drugs might simultaneously act on both pathogen and host. It would be interesting to determine if the effect on $T_{reg}$ cells correlates with clinical features of such drugs.

In order to develop improved diagnostic and therapeutic options, we will need a fuller understanding of the plethora of genes and pathways that regulate $T_{reg}$ cell differentiation and function. The majority of genetic polymorphisms that affect $T_{reg}$ cell function are unlikely to involve the few canonical genes that have been described. This issue will become increasingly pressing as genome sequencing technologies become more accessible and as our ability to manipulate immune modulation improves, requiring more precise selection of the right therapy for the right patient. Moreover, the identification of additional pathways will highlight new candidate therapeutic targets. It is important to note that these pathways, while ancillary to our current understanding, can be and likely are crucially important. This is underscored by our demonstration that $T_{reg}^{hi}$ conditions enhance $T_{reg}$ cell differentiation without increasing SMAD2 or SMAD3 phosphorylation above levels induced by $T_{reg}^{low}$ conditions. Thus, even the best understood $T_{reg}$-relevant cytokine, TGF-β1, engages signals

that remain to be fully understood. This notion is echoed by earlier discoveries that while ATRA enhances both SMAD3 signaling and $T_{reg}$ cell differentiation, the two are not linked as ATRA can enhance $T_{reg}$ cell differentiation in SMAD3-deficient mice (*Nolting et al., 2009*).

Although primary T cells have typically been less amenable to more traditional forward genetic approaches, we show here how chemical biology can rapidly advance our understanding of Th biology. Using a library enriched in tool compounds with known molecular activities enhanced our ability to rapidly make mechanistic insights. In this way, our discovery of harmine as a key compound of interest led us to uncover the novel activity of DYRK1A as a reciprocal regulator of $T_{reg}$ and Th17 cell differentiation.

The mechanistic details of how DYRK1A regulates Th differentiation remain to be clearly elucidated. Moreover, DYRK1A is a member of a family of five related proteins (*Aranda et al., 2011*). Whether other DYRK family members regulate $T_{reg}$ and Th17 cell differentiation will require additional experiments to elucidate. While the chemical inhibitors used are significantly more specific for DYRKs as compared to other families of kinases, their ability to distinguish between individual DYRKs is more limited (*Bain et al., 2007*; *Ogawa et al., 2010*; *Wippich et al., 2013*). The enhanced NFAT1 nuclear translocation we find associated with DYRK inhibition by harmine treatment in primary CD4+ T cells is consistent with previous studies in cell lines showing that DYRKs inhibit NFAT signaling (*Gwack et al., 2006*). On one hand, studies showing progressively severe defects in $pT_{reg}$ cell generation corresponding with increasing loss of NFAT family members raise the possibility that harmine-enhanced NFAT signaling may act in the opposite manner to promote $T_{reg}$ cell differentiation (*Vaeth et al., 2012*). Increased NFAT nuclear translocation might enhance binding to described FOXP3 enhancer elements, thus promoting FOXP3 expression, or even to FOXP3 itself, boosting transcription of FOXP3 targets (*Ruan et al., 2009*; *Wu et al., 2006*). However, in counterpoint to this simple association of increased NFAT with increased $T_{reg}$ cell differentiation, decreased NFAT signaling has also been reported to impair Th1 and Th17 cell differentiation (*Ghosh et al., 2010*; *Hermann-Kleiter and Baier, 2010*). Extrapolating these latter results, one might expect the harmine-enhanced NFAT signaling to concomitantly promote Th1 and Th17 cell differentiation, which we clearly do not observe. One way by which these conflicting predictions might be resolved could involve harmine modulating NFAT activity in a more complex manner, for example involving dynamics of nuclear retention, with a $T_{reg}$ cell-specific net effect. Alternatively, these results in conjunction with our finding that harmine's effect on nuclear localization of NFAT diminishes at later time points raise the possibility that some other target of DYRKs is more relevant in the context of $T_{reg}$ and Th17 cell differentiation.

Interestingly, human diseases secondary to perturbed DYRK function, on closer inspection, also exhibit immunological aspects, suggesting that the relationship between DYRKs and Th differentiation is physiologically germane. In this regard, the observation that DYRK inhibition promotes $T_{reg}$ cell differentiation leads to the converse prediction that increased DYRK activity would inhibit $T_{reg}$ cells. Down syndrome is characterized by trisomy of chromosome 21; specifically, the resulting increase in *DYRK1A* copy number and activity is thought to be a key driver of pathology (*Lepagnol-Bestel et al., 2009*). Notably, patients with Down syndrome have hypofunctional $T_{reg}$ cells and are at increased risk for autoimmune disease (*Pellegrini et al., 2012*). Similarly, gain-of-function mutations in *DYRK1B* were recently implicated in metabolic syndrome (*Keramati et al., 2014*). Decreased adipose tissue-associated $T_{reg}$ cells contribute to the inflammation that is a central player in obesity-induced metabolic syndrome (*Odegaard and Chawla, 2013*). It is tempting to speculate that our findings provide a unifying hypothesis that helps account for these disparate observations, with increased DYRK activity in these patients leading to decreased $T_{reg}$ cell differentiation via effects opposite to harmine's.

In addition to extending our understanding of the biology of $T_{reg}$ cell differentiation, our demonstration that harmine-enhanced $T_{reg}$ cells exhibit full functionality in multiple animal models of inflammation differing in genetic background, target organ system and antigen specificity raise interest in the possibility of manipulating this axis therapeutically. This notion is reinforced by our finding that harmine itself, directly administered, can attenuate inflammation. Furthermore, harmine similarly enhances human $T_{reg}$ differentiation, supporting the likely physiologic relevance of the pathways it engages. Taken together, we propose that DYRKs represent a novel, druggable target of particular relevance to tolerance and inflammation. In summary, these results illustrate how unbiased chemical biology approaches can identify novel chemical modulators of $T_{reg}$ cell differentiation, point to interesting mechanistic hypotheses and spark new translational efforts.

# Materials and methods

## Mice, antibodies and reagents

Balb/c, C57Bl/6, *Cd45.1*[+/+], *Rag1*[−/−], *Foxp3*[IRES-GFP], *Il17*[IRES-GFP], NOD-*scid* and NOD-*BDC2.5* mice were obtained from Jackson Labs. NOD-*BDC2.5.Foxp3*[IRES-GFP] mice were obtained from the JDRF Transgenic Core (Harvard Medical School, Boston, MA). *Bach2*-knockout mice have been previously described (*Muto et al., 2004*). Mixed chimeras were generated by injecting CD90.1[+]CD45.1[−]*Bach2*[−/−] and CD90.1[−]CD45.1[+]*Bach2*[+/+] bone marrow into C57Bl/6 hosts. Antibodies and cytokines used are described in *Supplementary file 1A*. Chemical compounds were sourced as in *Supplementary file 1B*. Pro-INDY and GSK-626616 were synthesized as previously described (*Corona et al., 2010*; *Ogawa et al., 2010*).

## Murine T cell isolation and culture

CD4[+]CD62L[+] naïve T cells were isolated using CD4 negative enrichment kits (Stemcell Technologies, Vancouver, Canada) and CD62L microbeads (Miltenyi Biotec, San Diego, CA) and confirmed to be >95% pure by flow cytometry. These were cultured on 96-well plates pre-coated with anti-CD3 and anti-CD28 under conditions outlined in *Supplementary file 2*. In particular, the addition of harmine to $T_{reg}{}^{low}$ conditions is abbreviated as $T_{reg}{}^{HAR}$. Compounds were pinned using a CyBIO CyBi Well Vario (96-well pintool) (Cybio, Jena, Germany). $T_{reg}$ and Th1 cultures were fed with equal volume of IL-2-supplemented media (10 ng/ml) and retreated with compound at day 2, split 1:2 into IL-2-supplemented media at day 3 and analyzed at day 4. Th17 and Th0 cultures were treated similarly except no IL-2 was supplemented. Cell proliferation was monitored using CFSE (Life Technologies, Carlsbad, CA) per manufacturer's instructions.

## Human T cell isolation and culture

Human peripheral mononuclear cells were separated using Ficoll–Paque (GE Healthcare, Little Chalfont, United Kingdom) and CD4[+]CD45RA[+] naïve T cells isolated using negative enrichment kits (Stemcell Technologies, Vancouver, Canada) per manufacturer's instructions and confirmed to be >90% pure by flow cytometry.

These were cultured on 96-well plates pre-coated with anti-CD3 and anti-CD28 under conditions outlined in *Supplementary file 2*. Cultures were fed with equal volume of IL-2-supplemented media (10 ng/ml) at day 4, split 1:2 into IL-2-supplemented media at day 6 and analyzed at day 8.

## Flow cytometry

5 hr prior to analysis, Th1 and Th17 cultures were restimulated with PMA and ionomycin (50 and 500 ng/ml respectively, Sigma Aldrich, St. Louis, MO) in the presence of Golgistop (BD Biosciences, San Jose, CA). Cells were typically stained with LIVE/DEAD and anti-CD4-FITC before being fixed and permeabilized using either Foxp3 fixation/permeabilization buffers (eBioscience, San Diego, CA) or Phosflow Fix/Perm buffers (BD Biosciences, San Jose, CA) as indicated. Intracellular staining was performed per manufacturer's instructions. Counting beads (10 μm, Spherotech, Lake Forest, IL) were added at 5000 per well. Acquisition was performed on a FACSVerse (BD Biosciences, San Jose, CA) and analyzed using FlowJo software (Treestar, Ashland, OR). Fractional enhancement was determined by increase in percentage lineage-committed cells, relative to maximal cytokine-driven enhancement. Fractional inhibition was calculated relative to percentage lineage-committed cells treated with DMSO. STAT3 phosphorylation was quantitated as previously described (*Chaudhry et al., 2011*). Cell sorting was performed on a FACSVantage (BD Biosciences, San Jose, CA).

## Screening data analysis and hit-calling

Each experimental 96-well plate included at least eight wells each of positive and negative controls. Each experimental batch included an additional plate of 48 positive and 48 negative controls and was processed separately. For quality control purposes, data from each experimental plate were first median-centered using data from all wells except positive controls. Median-centered data from all plates were pooled with batch-level negative controls to estimate batch-wide standard deviation. This step was repeated with the positive controls. Each plate was individually assessed if its internal controls met specific standards. Plates where ≤ 75% of controls scored within the expected range or

exhibited suboptimal dynamic range were excluded and retested subsequently. The remaining plates were subjected to a similar strategy of pooling median-centered data to estimate robust standard deviation. This measure was first used to select a negative control reference from the pool of in-plate negative controls and compound-treated wells. Next, data from each screening plate were transformed into Z-scores using the mean of select negative control wells and robust standard deviation. Z-normalized data from all screening plates were pooled per experimental batch. Generalized linear regression models were fitted to positive and negative controls using *glmfit* function in Matlab (Mathworks, Natick, MA). Compounds that performed at ≥30% of the observed levels for positive control (based on artemisinin's enhancement) were selected for secondary screening.

## Dose response curve fitting

Dose response curves for fractional enhancement of $T_{reg}$ cell differentiation and culture cellularity were analyzed in Matlab to identify $EC_{50}$ and $LD_{50}$ doses, at which 50% $T_{reg}$ enhancement and cytotoxicity are observed, respectively. Each compound was profiled across eight doses selected to sufficiently cover response dynamics. Dose response curves were fitted with either a single sigmoid or a double sigmoid function, depending on whether the response was asymptotic or impulse-like. An impulse function has the form:

$$f = \frac{1}{r_1} s(d : d_1, r_l, r_p, \alpha_1) \times s(d : d_2, r_h, r_p, \alpha_2),$$

where

$$s(d : d_m, r_i, r_f, \alpha) = r_i + (r_f - r_i) \frac{1}{1 + e^{-4\alpha(d - d_m)}},$$

is a sigmoid function with a response that ranges from $r_i$ to $r_f$ with mid-point at dose $d_m$ and a slope of $\alpha^*sign(r_f - r_i)$ at dose $d_m$. The parameters of this model describe the dose of response onset ($d_1$), dose of response offset ($d_2$), initial response at lowest dose ($r_l$), peak response ($r_p$), final response at highest dose ($r_h$), and two slope parameters to model the rate of response onset ($\alpha_1$) and offset ($\alpha_2$). A single-sigmoid function uses only four parameters ($d$, $r_1$, $r_2$, $\alpha$). All models were fitted to data using *fmincon* function in Matlab. Fitted models were reverse-queried to estimate the dose at which 50% of the peak response parameter was observed.

## Unsupervised clustering of phenotypic data

Phenotypic data ($LD_{50}/EC_{50}$ ratio, maximal $T_{reg}$ enhancement and Th1 and Th17 inhibition) for all 21 $T_{reg}$ enhancers were combined to form a feature matrix. The data were standardized and pairwise similarity between compounds was computed using Pearson correlation with complete linkage in GENE-E (http://www.broadinstitute.org/cancer/software/GENE-E/).

## Effects of chemical perturbation on gene expression

Transcriptomic profiles examining effects of compounds in three cell lines (MCF7, PC3 and HL60), available for 19 of 21 $T_{reg}$ enhancers, were downloaded from the Connectivity Map (CMAP) database and analyzed in Matlab (*Lamb et al., 2006*). Expression data from replicate experiments were averaged for each cell line; data from separate doses were not merged. A gene expression amplitude table of 22,280 genes and 62 CMAP instances (reduced from 151) was subject to principal components analysis for dimensionality reduction. 43 principal components explained up to 90% variance in expression data, using genes as features. Normalized PC scores for the first 43 components and 62 compound instances were analyzed for pairwise similarity using Euclidean distance with complete linkage in GENE-E.

## RNA isolation and qRT-PCR

RNA was isolated using RNeasy kits (Qiagen, Valencia, CA) and cDNA prepared using iScript cDNA synthesis kit (Bio-Rad, Hercules, CA) per manufacturer's instructions. Real-time PCR was performed using iTaq SYBR Green (Bio-Rad, Hercules, CA) on a C1000 thermal cycler (Bio-Rad, Hercules, CA) equipped with a CFX384 Real Time System (Bio-Rad, Hercules, CA). Cycling conditions were 95°C for 3 min, followed by 40 cycles of 94°C for 15 s, 59°C for 45 s, and 72°C for 6 s. Primers used were *Il17a*: TTTAACTCCCTTGGCGCAAAA and CTTTCCCTCCGCATTGACAC; *Il22*: CATGCAGGAGGTGGTACCTT

and CAGACGCAAGCATTTCTCAG; *Batf*: GACACAGAAAGCCGACACC and AGCACAGGGGCTCGTG; *Pou2af1*: CACCAAGGCCATACCAGGG and GAAGCAGAAACCTCCATGTCA; *Mina*: TTTGGGTCCTTA GTAGGCTCG and CCGATCCGGTCCTCAGATT; *Slc14a1*: GGCTCTGGGGTTTCAACA and GCCATC AGGTGTGCCATAC; *Trib1*: CAGATTGTTTCCGCCGTCG and ACCCTTAATGATGTGAGTATCTTCC; *Tfrc*: CCGCTCGTGGAGACTACTT and CCCAGAAGATATGTCGGAAAGG; *Mybl2*: CAAGAATGCCCTGGA GAAGTAC and GCTTTCTCTTCTGCTTCTCGG; *Mcm2*: GCCCATCATTTCCCGCTTTGA and CCCTTCA TCCTTCTTGTTACTGG; *Dab2ip*: CCATCCTCAGTGCCAAGAC and GGTCCACCTCTGACATCATCA; *Snx6*: GTTCTACAGGCTGAAACTTCCC and TAAAACCGCAAGGCAGTTCTG; *Exoc2*: GACAGCGTCAC TGAAGAGG and GAGTTTCCAGAAGTTAGGCAGC; *Slc9a8*: CTGGCAGAAGGAATCTCACTC and CAGTTCGGAGAGTCTGCTG; *Rora*: ATGCCACCTACTCCTGTCC and GACATCCGACCAAACTTGACAG; *Akt3*: GGCACACCAGAGTACCTG and GCATCTGAAGAGAGTGTTCGG; *Lmna*: TGTGGCGGTAG AGGAAGT and GGAAGCGATAGGTCATCAAAGG; *St3gal3*: GACTGCCATCTTCCCCAG and CAAAAGGTGGCACAAACTCC; *Ccl20*: TACTGCTGGCTCACCTCTG and CCATCTGTCTTGTGAAACC CAC; *Aqp3*: TTGGCATCTTGGTGGCTG and GCTCATTGTTGGCAAAGGC; *Ypel5*: CCAATCGCTCA GAACTCATCTC and ATAACGCTGGCTGTCTTCAG.

## Gene expression profiling

RNA concentration and purity were measured using a NanoDrop spectrophotometer (Thermo Scientific, Waltham, MA). For microarray studies, RNA was amplified and labeled using the Illumina TotalPrep RNA Amplification kit (Ambion, Grand Island, NY) per manufacturer's instructions. Labeled cRNA was then hybridized to the Illumina Mouse WG-6 v2 chip. The Illumina microarray was performed by Partners HealthCare Center for Personalized Genetic Medicine (PCPGM) Translational Genomics Core (Boston, MA). BeadChips were scanned per protocol (Illumina Whole Genome Gene Expression for BeadStation Manual v3.2, Revision A) using scanning software BeadScan 3.5.31. The GenomeStudio Data Analysis Software (Illumina, San Diego, CA) was used for data collection. The final report comprising the full dataset was initially processed using the Bioconductor package Lumi by employing a background correction estimate. Subsequently, signal intensities were VST-transformed (variance-stabilizing transformation) and RSN-normalized (robust spline normalization) using the Lumi package in R. Post-processing and statistical analysis of microarray data was carried out in Matlab. Normalized $\log_2$ data was first checked for correlation between replicates (>0.98 on average) and probes without gene assignments removed.

For RNAseq studies, polyA mRNA was isolated using the Dynabeads mRNA DIRECT kit (Life Technologies, Grand Island, NY) and cDNA generated using poly-dT priming and Maxima reverse transcriptase (Life Technologies, Grand Island, NY) per manufacturer's instructions. After RNase A (Life Technologies, Grand Island, NY) treatment and SPRI bead (Beckman Coulter, Pasadena, CA) cleanup, second strand cDNA was synthesized using NEBNext mRNA Second Strand Synthesis Module (New England Biolabs, Ipswich, MA) followed by SPRI cleanup per manufacturer's instructions. Samples were tagmented using Nextera DNA sample preparation kits (Illumina, San Diego, CA) and read on a MiSeq (Illumina, San Diego, CA) per manufacturer's instructions. RNAseq reads were aligned using Tophat (mm10) and RSEM-based quantification using known transcripts (mm10) performed, followed by further processing using the Bioconductor package DESeq in R. The data was normalized by library size followed by variance estimation and evaluation of pairwise differential expression based on a negative binomial distribution model. Post-processing and statistical analysis was carried out in Matlab. The data discussed in this publication have been deposited in NCBI's Gene Expression Omnibus and are accessible through GEO Series accession number GSE67961 (http://www.ncbi.nlm.nih.gov/geo/query/acc.cgi?acc=GSE67961) (*Edgar et al., 2002*).

Gene signatures overlaid were previously reported and $\chi^2$ analyses were performed as previously reported (*Hill et al., 2007*; *Zheng et al., 2009*; *Darce et al., 2012*; *Burzyn et al., 2013*; *Roychoudhuri et al., 2013*; *Joller et al., 2014*). To identify the $T_{reg}^{HAR}$ signature, we compared the 2269 genes (Student's t-test, nominal $p \leq 0.05$) differentially regulated between sorted $T_{reg}^{hi}$ and $T_{reg}^{HAR}$ cells with the 3492 differentially expressed genes (Student's t-test, nominal $p \leq 0.05$) between human $T_{reg}$ and Th17 cells (*Zhang et al., 2013*). Gene expression from each experiment was independently scaled on a per-gene basis using min–max normalization, analyzed in GENE-E and marker selection used to identify the 111-gene signature with signal-to-noise ratio > 1. Transcription factor binding site prediction and enrichment was performed using the Molecular Signatures Database (MSigDB) (*Subramanian et al., 2005*).

To assess enrichment in disease-associated genes, we downloaded the 1437 genes previously reported to lie within IBD-associated loci (*Jostins et al., 2012*). We used permutation testing as previously reported to assess fold-enrichment and significance of overlap between the $T_{reg}^{HAR}$ signature set and the IBD-associated gene set (*Okada et al., 2014*). Briefly, for each of 10,000 iterations, we randomly selected a set of 1437 genes from the entire genome, assessing for overlap with the $T_{reg}^{HAR}$ signature gene set. This distribution was used as the background to assess fold enrichment. Significance of the enrichment was evaluated using one-sided permutation tests.

## Lentiviral knock-down

Lentiviral shRNA constructs were designed based on established TRC guidelines and cloned at the Genetic Perturbation Platform (Broad Institute, Cambridge, MA). Sequences used to knock-down *Dyrk1a* expression were TATGAAATCGACTCCTTAATA, TTTGAAATGCTGTCCTATAAT, GAGGTC GATCAGATGAATAAA, GAACCCGTAAACTTCATAATA and ACTCGGATTCAACCTTATTAT. Non-targeting sequences used were AGCAGCTGTTCGAGGATAATA, TTTGCACAAGAACAGAATAAT and ACAGATGCCAATGGGAATATT. Naïve CD4$^+$ T cells were isolated and stimulated as described above in Th0 conditions. Cells were infected with 15 μl concentrated lentiviral supernatant at day 1, fed with media + polarizing cytokines at day 2 ($T_{reg}^{low}$ or Th17$^{hi}$ conditions as described above), split 1: 2 at day 4 and analyzed at day 5. For RNAseq studies, cells were fed with media $\pm$ harmine at day 2 without polarizing cytokines and harvested at day 4.

## Quantitative imaging flow cytometry

Sorted CD4$^+$ cells were stimulated as indicated, fixed in 3% paraformaldehyde (Santa Cruz, Dallas, TX), permeabilized in PBS + 2% fetal calf serum + 0.1% Triton X-100 (Sigma Aldrich, St. Louis, MO) and stained with anti-CD4-FITC, rabbit anti-NFAT1, APC-donkey anti-rabbit and DAPI. Acquisition was performed on an Amnis ImageStream MkII imaging flow cytometer (EMD Millipore, Billerica, MA), combining high-resolution microscopy and flow cytometry to quantitate nuclear/cytoplasmic signal distribution. 10,000 event data files were acquired per sample using a 60× objective running a 6-μm core diameter at 44 mm/s. Brightfield, side scatter and fluorescent images were collected using three excitation lasers (405 nm, 1 mW Ch07; 642 nm, 100 mW Ch11; 785 nm, 1.09 mW for SSC Ch06). Single color controls were acquired and spectral compensation performed post-acquisition. From single, nucleated cells in focus, the DAPI$^+$CD4$^+$NFAT$^+$ population was gated upon. To determine NFAT1 signal distribution, a nuclear morphology mask, which includes all pixels within the outermost image contour, was created from the DAPI$^+$ image (Channel 07). Nuclear/whole cell NFAT1 ratios were determined by dividing the NFAT1 intensity within the defined nuclear morphology mask by the NFAT1 intensity over the entire cell (default MC Ch11 mask). Nuclear/cytoplasmic histogram values were compared using a similarity score (a log-transformed Pearson's correlation coefficient that measures the degree to which two images are linearly correlated in the nuclear masked region).

## Western blotting and nuclear isolation

Cell extracts for Western analyses were prepared using TNN lysis buffer, pH 8 (100 mM TRIS-HCl, 100 mM NaCl, 1% NP-40, 1 mM DTT, 10 mM NaF) supplemented with protease inhibitor (Roche, Indianapolis, IN) and phosphatase inhibitors (Thermo Scientific, Waltham, MA), separated by SDS-PAGE (Bio-Rad, 456-9035) and transferred onto PVDF membrane (Immobilon-P, Millipore, IPVH20200, Billerica, MA) Approximately 10$^6$ cells were processed per lane. Antibodies used are described in *Supplementary file 1A*. Bands were visualized using Western Lightning Plus-ECL (Perkin Elmer, Waltham, MA) and/or SuperSignal West Femto substrate (Thermo Scientific, Waltham, MA) per manufacturer's instructions. Nuclear isolation was performed using a Nuclei Isolation Kit (Sigma Aldrich, St. Louis, MO) per manufacturer's protocol after cells were lysed in RIPA buffer (150 mM NaCl, 1% Triton X-100, 0.5% sodium deoxycholate, 0.1% SDS, 50 mM TRIS-HCl pH7.8) supplemented with DTT, protease and phosphatase inhibitors. Band intensity was quantitated using ImageJ (*Schneider et al., 2012*).

## In vitro T$_{reg}$ suppression assay

This was performed as previously described (*Collison and Vignali, 2011*). Briefly, sorted CD45.1$^+$CD4$^+$CD62L$^+$ T$_{responder}$ cells were labeled with CFSE (Invitrogen, Grand Island, NY) per manufacturer's protocol, plated at 5 x 10$^4$ cells per well in 96-well U bottom plates and co-cultured

with sorted CD45.2$^+$Foxp3$^{IRES-GFP}$ T$_{reg}$ cells at ratios indicated in the presence of anti-CD3/28 beads (Dynabead, Grand Island, NY) and analyzed by flow cytometry 3 days later.

## T$_{reg}$ suppression—T1D model

As previously described, $5 \times 10^4$ CD4$^+$CD62L$^+$ T cells isolated from NOD-*BDC2.5* mice were administered intravenously to NOD-*scid* mice with or without $1 \times 10^5$ sorted T$_{reg}$ cells generated from NOD-*BDC2.5.Foxp3* $^{IRES-GFP}$ mice (*Herman et al., 2004*; *Tarbell et al., 2004*). Blood glucose levels were measured with a handheld Contour glucometer (Bayer, Leverkusen, Germany) at days 3, 6, 8 and every day thereafter. Diabetes was diagnosed when blood sugar was over 250 mg/dl for 2 consecutive days.

## T$_{reg}$ suppression—CD45RB$^{hi}$ colitis model

As previously described, $5 \times 10^5$ CD4$^+$CD62L$^+$ T cells were injected into the intraperitoneal cavity of *Rag1*$^{-/-}$ mice. 10 days later, mice were injected with either PBS or $1.5 \times 10^5$ sorted T$_{reg}$ cells generated from *Foxp3* $^{IRES-GFP}$ mice (*Smith et al., 2013*). Mice were monitored weekly for weight loss and morbidity for 6–8 weeks per protocol. At 8 weeks, mice were euthanized and proximal, middle and distal colon analyzed histologically by blinded observers as previously described (*De Jong et al., 2000*).

## T$_{reg}$ suppression—airway inflammation model

Allergic airway inflammation was induced in mice as previously described (*Grainger et al., 2010*). In brief, C57Bl/6 mice were injected intraperitoneally with 10 µg of ovalbumin (Sigma–Aldrich, St. Louis, MO) and 1 mg of aluminum hydroxide (Sigma–Aldrich, St. Louis, MO) suspended in 0.5 ml of PBS on days 0 and 10. Sorted T$_{reg}$ cells generated from *Foxp3*$^{IRES-GFP}$ mice using T$_{reg}$$^{hi}$/T$_{reg}$$^{HAR}$ conditions were transferred by retroorbital injection on days 16 and 19. Mice were challenged intratracheally with 10 µg OVA in PBS on days 17 and 20 and sacrificed 20–24 hr after the last challenge. The trachea was exposed and cannulated with polyethylene tubing followed by bronchoalveolar lavage (BAL) with PBS + 0.6 mM EDTA. Lavage fluid was centrifuged, and pelleted cells counted and analyzed. The differential cell count was performed as previously described; cells were stained with Diff-Quick (Dade Behring, Newark, DE) and macrophages, neutrophils, eosinophils, and lymphocytes on cytocentrifuge preparations enumerated (*Grainger et al., 2010*). At least 200 cells were counted on each slide.

## Protection against airway inflammation

C57Bl/6 mice were injected intraperitoneally with 100 µg ovalbumin and 1 mg aluminum hydroxide on day 0, challenged with 10 µg ovalbumin intratracheally on days 14, 17 and 21, and tissues were harvested for analysis 24 hr after the last challenge as previously described (*Curotto de Lafaille et al., 2008*). Tolerance was induced by intranasal sensitization with 100 µg ovalbumin prior to immunization on days −7, −6, and −5 (*Curotto de Lafaille et al., 2008*). Mice were treated daily with 1 mg harmine HCl (Santa Cruz Biotechnology, Dallas, TX) dissolved in water intranasally from days −8 through −3 as indicated.

## Histology

Tissues were preserved in 10% formalin. Paraffin embedding, sectioning and staining with either hematoxylin and eosin or Periodic acid-Schiff/Alcian Blue were performed by the Histopathology Research Core (Massachusetts General Hospital, Boston, MA)

## Effects of harmine in vivo

Mice were treated with 1 mg harmine HCl intranasally for 5 days, and thoracic lymph nodes harvested on day 6 (for T cell studies) or day 2 (for dendritic cell studies). To liberate dendritic cells, lymph nodes were mechanically disrupted and incubated in HBSS (GE Healthcare, Little Chalfont, United Kingdom) containing 2.5 mg/ml collagenase D (Roche Diagnostics, Indianapolis, IN) at 37°C for 30 min. Digestion was neutralized by adding EDTA to 20 mM.

## Statistical analyses

Statistical measures, including mean values, standard deviations, Student's t-tests, Mantel–Cox tests, Mann–Whitney tests and one-way ANOVA tests, were performed using Graphpad Prism software and Matlab. Where appropriate, unless otherwise stated, graphs display mean ± standard deviation.

## Study approval

All experiments were performed with the approval of the IACUC of Massachusetts General Hospital (Boston, MA).

## Acknowledgements

We thank Paul Clemons, Kara Lassen, Nicole Desch and Natalia Nedelsky for helpful discussion and Isabel Latorre for assistance with project management. BK was supported by N.I.H. grants T32 CA009216-31, T32 HL066987-12 and K08 DK104021-01. Work was funded by N.I.H. grant S10 OD012027-01A1 to Frederic I. Preffer at Massachusetts General Hospital. RJX was supported by the Crohn's and Colitis Foundation of America, the Leona M and Harry B. Helmsley Charitable Trust and N.I.H. grants P30 DK043351 and U01 DK062432. NPR was supported by the Intramural Research Program of the National Cancer Institute. RR was supported by a Sir Henry Dale Fellowship jointly funded by the Wellcome Trust and the Royal Society (Grant Number 105663/Z/14/Z).

## Additional information

### Funding

| Funder | Grant reference | Author |
| --- | --- | --- |
| National Institutes of Health (NIH) | T32 CA009216-31 | Bernard Khor |
| National Institutes of Health (NIH) | T32 HL066987-12 | Bernard Khor |
| National Institutes of Health (NIH) | K08 DK104021-01 | Bernard Khor |
| National Institutes of Health (NIH) | S10 OD012027-01A1 | Scott Mordecai |
| Crohn's and Colitis Foundation of America (CCFA) | | Ramnik J Xavier |
| Leona M. and Harry B. Helmsley Charitable Trust | | Ramnik J Xavier |
| National Institutes of Health (NIH) | P30 DK043351 | Ramnik J Xavier |
| National Institutes of Health (NIH) | U01 DK062432 | Ramnik J Xavier |
| National Cancer Institute (NCI) | Intramural Research Program | Nicholas P Restifo |
| Wellcome Trust | Sir Henry Dale Fellowship | Rahul Roychoudhuri |
| Royal Society | Sir Henry Dale Fellowship, Grant number 105663/Z/14/Z | Rahul Roychoudhuri |

The funders had no role in study design, data collection and interpretation, or the decision to submit the work for publication.

### Author contributions

BK, Conception and design, Acquisition of data, Analysis and interpretation of data, Drafting or revising the article; JDG, GG, MIR, KLC, LNA, AMP, SM, Acquisition of data, Analysis and interpretation of data, Drafting or revising the article; KT, TBS, DD, PHT, Acquisition of data, Drafting or revising the article; MS, AKB, BDM, SLS, AHS, SYS, Analysis and interpretation of data, Drafting or revising the article; RR, Conception and design, Drafting or revising the article, Contributed unpublished essential data or reagents; NPR, JJO'S, Drafting or revising the article, Contributed unpublished essential data or reagents; AFS, RJX, Conception and design, Analysis and interpretation of data, Drafting or revising the article

### Ethics

Human subjects: This research involved the collection and study of deidentified existing specimens. Animal experimentation: This study was performed in strict accordance with the recommendations in the Guide for the Care and Use of Laboratory Animals of the National Institutes of Health. All of

the animals were handled according to approved institutional animal care and use committee (IACUC) protocols (#2007N000126) of the Massachusetts General Hospital. The protocol was approved by the IACUC Committee of the Massachusetts General Hospital (OLAW File Number: A3596-01).

## Additional files

### Supplementary files

• Supplementary file 1. (**A**) Staining reagents used. (**B**) Chemicals used.

• Supplementary file 2. Th conditions used.

### Major datasets

The following dataset was generated:

| Author(s) | Year | Dataset title | Dataset ID and/or URL | Database, license, and accessibility information |
| --- | --- | --- | --- | --- |
| Khor B, Goel G, Xavier RJ | 2014 | The kinase DYRK1A reciprocally regulates the differentiation of Th17 and T regulatory cells | http://www.ncbi.nlm.nih.gov/geo/query/acc.cgi?acc=GSE67961 | Publicly available at NCBI Gene Expression Omnibus (Accession No. GSE67961). |

Standard used to collect data: NCBI GEO submission is MIAME-compliant.
The following previously published datasets were used:

| Author(s) | Year | Dataset title | Dataset ID and/or URL | Database, license, and accessibility information |
| --- | --- | --- | --- | --- |
| Lamb J, Crawford ED, Peck D, Modell JW, Blat IC, Wrobel MJ, Lerner J, Brunet JP, Subramanian A, Ross KN, Reich M, Hieronymus H, Wei G, Armstrong SA, Haggarty SJ, Clemons PA, Wei R, Carr SA, Lander ES, Golub TR | 2006 | Connectivity Map dataset (build01) | http://www.ncbi.nlm.nih.gov/geo/query/acc.cgi?acc=GSE5258 | Publicly available at NCBI Gene Expression Omnibus (Accession No. GSE5258). |
| Zhang H, Nestor CE, Zhao S, Lentini A, Bohle B, Benson M, Wang H | 2013 | Profiling of human CD4+ T subsets identifies a Th2-specific non-coding RNA GATA3-AS1 | http://www.ncbi.nlm.nih.gov/geo/query/acc.cgi?acc=GSE43005 | Publicly available at NCBI Gene Expression Omnibus (Accession No. GSE43005). |

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
