## [Decision Letter]

Thank you for sending your work entitled “DYRK1a is a novel, reciprocal regulator of T_reg_ and Th17 cell differentiation” for consideration at *eLife*. Your article has been favorably evaluated by Charles Sawyers (Senior editor), a Reviewing editor, and three reviewers.

The Reviewing editor and the reviewers discussed their comments before we reached this decision, and the Reviewing editor has assembled the following comments to help you prepare a revised submission.

This study aims to address an unmet need, namely to develop drugs that specifically enhance T_regs_ while suppressing Th1 and Th17. Such agents are needed for autoimmune and autoinflammatory diseases, for which current treatment options are limited. This study combines several state of the art approaches, including chemoinformatics, bioinformatics, and an experimental system involving primary CD4 T cells to work toward this goal. Using an already existing drug library, they identified a panel of compounds that affect T_reg_/Th17 differentiation in vitro. They then studied one of these compounds (Harmine) that inhibited a kinase DYRKi1, and enhanced T_regs_ while suppressing Th17 development. The authors also showed that treatment with Harmine inhibited inflammation in multiple disease models.

The reviewers were impressed with the approach employed by the authors to screen for compounds. But, to rise to the level of a paper in *eLife*, several significant concerns need to be addressed. This is partly because one could argue that the identification of Harmine by itself could have been deduced based on past knowledge. Harmine has been reported to regulate NFAT activation in osteoblast through inhibition of DYRK mediated NFAT phosphorylation and inactivation (Egusa et al., Bone, 2011). Extrathymic induction of Foxp3 and T_reg_ generation has been shown to be dependent on NFAT (Vaeth et al., PNAS, 2012). The reviewers have suggested that a number of important points need to be addressed, which are summarized below.

Major comments:

1) Additional evidence is required to establish the biological mechanism by which Harmine functions to enhance T_regs_. In particular, a more rigorous mechanistic examination of the role of DYRK kinases and their downstream targets using genetic approaches is needed. For example, addressing the following questions may achieve a deeper mechanistic understanding:

Is DYRK kinase activity selectively modulated during Foxp3 vs. Th17 induction, and if so, how?

Does such a modulation affect generation of T_reg_ vs. Th17 cells in vivo?

What are the differences in genetic/epigenetic signatures between DYRK-deficient systems and treatment with Harmine?

What is the role of *Bach2*-suppression in the effects of Harmine?

2) The effects of Harmine on T_reg_ expansion versus relative toxicity to other cell types need to be characterized more comprehensively. Addressing the following questions might achieve this goal:

Does Harmine administration in vivo affect absolute numbers of T_reg_ and effector T cells, or antigen presenting cell subsets? NFAT activity can modulate properties of antigen-presenting cells.

Do effects of Harmine on these cell subsets contribute to the observed therapeutic effects in vivo?

Does Harmine treatment in vitro affect total cell number (Foxp3^+^, IL-17^+^, and overall cell yield) and cell division?

3) It is important to describe the effect of Harmine on human cells. An in vitro study is necessary for this paper. A human pilot study is *not* necessary, even though the drug is approved for use.

---

## [Author Response]

*1) Additional evidence is required to establish the biological mechanism by which Harmine functions to enhance T*_*regs*_*. In particular, a more rigorous mechanistic examination of the role of DYRK kinases and their downstream targets using genetic approaches is needed. For example, addressing the following questions may achieve a deeper mechanistic understanding*:

*Is DYRK kinase activity selectively modulated during Foxp3 vs. Th17 induction, and if so*, *how?*

DYRK proteins are synthesized as constitutively active kinases (Aranda et al., FASEB J 2011;25:449). We examined levels of DYRK1A protein in both T_reg_ and Th17 cells by FACS and Western blot. DYRK1A levels were consistently lower in T_reg_ cells as compared to Th17 cells (new Figure 4). Importantly, Th17-polarizing conditions (TGFβ + IL-6 + IL-1β) typically induce IL-17 expression in about 40% of cells; the remaining cells that did not become Th17 cells expressed DYRK1A at levels comparable to T_reg_ cells, further supporting that this difference in DYRK1A is specific to T_reg_ vs Th17 cells (new Figure 4). Together, these results extend our findings and demonstrate that DYRK1A represents a physiologically relevant modifier pathway that reciprocally modulates T_reg_/Th17 differentiation.

*Does such a modulation affect generation of T*_*reg*_
*vs. Th17 cells in vivo?*

We have added shRNA studies demonstrating that knockdown of *Dyrk1a* expression promotes T_reg_ differentiation and inhibits Th17 differentiation in vitro, in accordance with our observed effects of harmine (new Figure 4). These results correspond with our in vivo results. Here, we administer the DYRK1A inhibitor harmine to mice intranasally, and find an increase in T_reg_s in the local thoracic lymph nodes (Figure 2, new Figure 2—figure supplement 1). No appreciable effect on antigen-presenting cells was observed (new Figure 2—figure supplement 2). Furthermore, we demonstrate functional consequences of this difference, with intranasal harmine treatment powerfully attenuating recall airway inflammation in an experimental model of asthma (Figure 2). These data support the notion that modulating DYRK activity affects T_reg_ generation in vivo.

Our attempts to demonstrate similar effects using genetic methods to manipulate DYRK1A levels have been complicated for technical reasons. The *Dyrk1a* knockout is lethal, and the T cell conditional knockout exhibits significant defects in early T cell development, confounding studies late in development. We explored the potential utility of the Ts65Dn model, which is trisomic for approximately two-thirds of the genes on human chromosome 21 (including *Dyrk1a*). Although this model is used in neurological studies of Down syndrome, we found that DYRK1A levels in lymphoid populations (including CD4^+^CD62L^+^ T cells, CD4^+^CD44^+^ T cells, CD8^+^ T cells, and B cells) were no higher in Ts65Dn *Dyrk1a* trisomic cells than in wild type controls, suggesting that this may not be a good model to study DYRK1A dysregulation-related events late in CD4^+^ ontogeny. Consistent with this notion, we did not find any differences in T_reg_/Th17 populations in the spleen, lymph nodes, or lamina propria either at baseline or after induction of colitis using TNBS (a Th17-relevant model).

Our attempts to extend the shRNA knockdown experiments into in vivo models have also been confounded by technical limitations. Here, we activated cells with anti-CD3/28 for 24 hours and transduced them with shRNA against either *Dyrk1a* or control. 24 hours later, these cells were transferred into hosts; aliquots maintained in culture showed >90% infection rate. When transferred into RAG-deficient hosts and examined 5 days later, we could not recover any adoptively transferred cells, whether transduced with control or *Dyrk1a* shRNA. Additionally, we transferred transduced OT2 cells into B6 hosts and transduced DO11 cells into Balb hosts concomitant with intranasal OVA/curdlan immunization and analyzed thoracic lymph nodes 2 days later. Again, we were unable to recover adoptively transferred cells. These methods have been successfully applied using CD4^+^ T cells that were not pre-activated. However, activation is a requisite step to permit transduction, and genetic models with altered DYRK1A levels are not readily available. Taken together, we believe that the intranasal harmine data represent the best feasible in vivo experiment at this time.

What are the differences in genetic/epigenetic signatures between DYRK-deficient systems and treatment with Harmine?

This excellent suggestion led us to perform RNAseq studies. To maintain cell-context relevance, we performed these studies in primary CD4^+^ T cells, which we stimulated under Th0 conditions (in part to prevent confounding differences due to different levels of T_reg_ differentiation) followed by transduction with control- or *Dyrk1a*-shRNA. Half of the control-shRNA-transduced cells were treated with harmine. These results revealed significant similarity between *Dyrk1a* knockdown and harmine treatment (new Figure 4—figure supplement 2, Pearson correlation = 0.66, χ^2^ tests of up-/down-regulated genes *P*=0). Overall, our data are consistent with the notion that DYRK1A inhibition represents a major mechanism by which harmine regulates T_reg_/Th17 differentiation. In the setting of these results, we reasoned that epigenetic studies, while also potentially interesting, might be of lower priority at this time.

What is the role of Bach2-suppression in the effects of Harmine?

This interesting question led us to collaborate with Drs. O’Shea and Restifo to examine the effect of harmine in *Bach2*-knockout cells. We were able to obtain chimeras and stimulate BACH2-sufficient and -deficient cells in the same well, thus addressing the intricacies of culture cellularity affecting T_reg_ differentiation in low TGFβ conditions, as well as providing an internal control for effect of harmine. These studies recapitulate the cell-intrinsic and cell-extrinsic defects associated with BACH2 deficiency. Importantly, we show that harmine can enhance T_reg_ and inhibit Th17 differentiation in BACH2-deficient cells, demonstrating that harmine can act on BACH2-independent pathways and suggesting therapeutic relevance in BACH2-related inflammatory disease (new Figure 4). Interestingly, we also observe that harmine cannot rescue the differentiation of BACH2-deficient cells to the levels of wild type cells, consistent with the notion that harmine also engages BACH2-related pathways (new Figure 4).

*2) The effects of Harmine on T*_*reg*_
*expansion versus relative toxicity to other cell types need to be characterized more comprehensively. Addressing the following questions might achieve this goal*:

*Does Harmine administration in vivo affect absolute numbers of T*_*reg*_
*and effector T cells, or antigen presenting cell subsets? NFAT activity can modulate properties of antigen-presenting cells*.

This question led to a closer examination of the effects of harmine administration in vivo. Our results demonstrate that early after treatment, harmine increases the cellularity of the thoracic lymph node approximately two-fold. Importantly, there is a specific and reproducible increase in T_reg_ percentage and absolute numbers, although only the former remains statistically significant after correction for multiple comparisons. The other CD4^+^ naïve and effector populations are generally elevated by 50-100%, although this doesn’t reach statistical significance (new Figure 2—figure supplement 1).

With regards to antigen-presenting cells, we examined absolute numbers and expression of several costimulatory molecules known to modulate T_reg_ generation (CD40, CD80, and CD86). In both classical and migratory dendritic cells, these parameters all appeared to be unaffected by harmine administration (new Figure 2—figure supplement 2).

Do effects of Harmine on these cell subsets contribute to the observed therapeutic effects in vivo?

Given the results described above, the effect of harmine on T_reg_ generation via direct effects on CD4^+^ T cells appears to be the dominant contributor to the therapeutic effects observed in vivo.

*Does Harmine treatment in vitro affect total cell number (Foxp3*^*+*^, *IL-17*^*+*^*, and overall cell yield) and cell division?*

This is a good question that led us to look more closely at the effect of harmine on absolute cell counts and division in comparison to known T_reg_ enhancers, which we believe will be of general interest and value. At the dose tested, harmine does not exhibit a significant effect on total cellularity in vitro, with the consequence that absolute T_reg_ numbers are elevated to levels comparable to that of high TGFβ, and significantly higher than levels achieved by rapamycin (new Figure 3). Along these lines, both percentage and absolute numbers of Th17 cells are significantly reduced by treatment with harmine (new Figure 3—figure supplement 2). CFSE studies suggest at most a modest delay in proliferation at earlier time points (e.g., 24% at day 3) with harmine treatment although this catches up by day 4 (new Figure 3—figure supplement 1). In comparison, rapamycin causes a 60% reduction in proliferating cells at day 3, with a persistent 15% reduction at day 4 (new Figure 3—figure supplement 1).

*3) It is important to describe the effect of Harmine on human cells. An in vitro study is necessary for this paper. A human pilot study is* not *necessary, even though the drug is approved for use*.

This is an excellent comment that led us to demonstrate that harmine also potently enhances T_reg_ differentiation in human CD4^+^ T cells (new Figure 3), even more so than high TGFβ conditions, further suggesting physiologic and therapeutic relevance of pathways engaged by harmine.